# MEMORY-AUGMENTED TRANSFORMERS CAN IMPLEMENT LINEAR FIRST-ORDER OPTIMIZATION METHODS

## ABSTRACT

We show that memory-augmented Transformers (**Memformers**) can implement linear first-order optimization methods such as conjugate gradient descent, momentum methods, and more generally, methods that linearly combine past gradients. Building on prior work that demonstrates how Transformers can simulate preconditioned gradient descent, we provide theoretical and empirical evidence that Memformers can learn more advanced optimization algorithms. Specifically, we analyze how memory registers in Memformers store suitable intermediate attention values allowing them to implement algorithms such as conjugate gradient. Our results show that Memformers can efficiently learn these methods by training on random linear regression tasks, even learning methods that outperform conjugate gradient. This work extends our knowledge about the algorithmic capabilities of Transformers, showing how they can learn complex optimization methods.

## 1 INTRODUCTION

In-context learning (ICL) allows large language models (LLMs) to generate contextually appropriate outputs based solely on examples and queries provided in a prompt, without requiring any parameter adjustments (Brown, 2020; Liu et al., 2021; Lu et al., 2021; Wei et al., 2022; Wu et al., 2022). This remarkable ability has spurred research into understanding how Transformers can implement algorithms (Achiam et al., 2023; Touvron et al., 2023), with recent studies focusing on their capability to simulate optimization algorithms (Dai et al., 2022; Von Oswald et al., 2023a; Garg et al., 2022; Akyürek et al., 2022). Transformers have been shown to implement gradient-based optimization during their forward pass, such as preconditioned gradient descent for linear regression tasks (Dai et al., 2022; Mahankali et al., 2023; Ahn et al., 2024).

More recently, studies have demonstrated that Transformers can learn even more advanced optimization methods. For instance, Fu et al. (2023) showed that Transformers exhibit convergence rates comparable to Iterative Newton's Method, a higher-order optimization technique that converges exponentially faster than gradient descent for in-context linear regression. Additionally, Vladymyrov et al. (2024) proved that Transformers can, in fact, learn a variant of gradient descent that approximates second-order methods, such as $GD^{++}$, achieving convergence rates similar to Newton's method. These findings lead to the central question of our paper:

*Can Transformers efficiently "learn" more advanced gradient-based optimization methods?*

We aim to address this question by revealing some of the representational power of Transformers as "algorithm learners," further motivating the use of machine learning for discovering new optimization algorithms. To make our investigation more precise, we focus on learning the class of gradient-based algorithms obtained by linearly combining past gradients, known as *Linear First-Order Methods (LFOMs)* (Goh, 2017), where the $(k + 1)$st iterate is

$$w^{k+1} = w^0 + \sum_{i=0}^{k} \Gamma_i^k \nabla f(w^i), \tag{1}$$

and where $\{\Gamma_i^k\}_{i=0}^k$ are diagonal matrices. Model (1) is quite general, as it includes, as special cases, standard methods such as gradient descent (GD), momentum GD, Nesterov's accelerated gradient, conjugate gradient, and in a stochastic setting, AdaGrad, ADAM, among others.

By **"learning"** an algorithm like CGD or LFOM, we mean two key things:

1. *The Memformer, in its forward pass, under certain internal parameter settings, can perform iterations of CGD and/or LFOM.* This means that its architecture and parameterization are sufficiently expressive to execute these optimization methods as part of its computation.

2. *The Memformer's learnable parameters can be trained on linear regression tasks. When using these learned parameters, which are shared across all in-context data samples in a batch, the Memformer can execute "CGD-like" and "LFOM-like" iterations during a forward pass.* The surprising aspect lies in the Memformer's ability to achieve competitive—and in some cases even superior—performance compared to CGD, despite using a relatively small number of learned parameters shared across all test samples drawn independently of the training data.

Our key insight for efficiently learning LFOMs is to leverage memory-augmented Transformers, known as *Memformers* (Wu et al., 2020; Xu et al., 2021), which retain intermediate attention values across layers. This memory enables Memformers to store past gradients, facilitating the execution of advanced first-order methods such as conjugate gradient descent and momentum methods. The same mechanism allows Memformers to implement more general LFOMs.

While unconditional learning of gradient methods remains out of reach, we build on related work demonstrating that Transformers can learn gradient descent in the context of linear regression tasks (Garg et al., 2022; Akyürek et al., 2022; Von Oswald et al., 2023a; Ahn et al., 2024; Zhang et al., 2024). Inspired by these findings, and extending the work of Ahn et al. (2024), we conduct a theoretical analysis of the loss landscape for memory-augmented linear Transformers that omit softmax activation (Schlag et al., 2021; Von Oswald et al., 2023a; Ahn et al., 2024).

In the Appendix, we also include our experiments that Memformers can outperform Nesterov Accelerated Gradient (NAG) and momentum GD. In summary, our main contributions are as follows:

MAIN CONTRIBUTIONS

(1) **Theoretical justification that Memformers can implement LFOM iterations, including CGD.** We provide a rigorous theoretical framework showing that Memformers, when trained on linear regression tasks, can be configured to perform iterations of LFOMs in their forward pass, encompassing advanced algorithms like CGD. By leveraging their memory mechanisms, Memformers can store and effectively combine past gradients, enabling them to implement these sophisticated optimization methods within their architecture.

(2) **Empirical evidence of Memformers "learning" optimization algorithms.** Through extensive experiments, we demonstrate that Memformers can learn LFOMs, in a general sense, by training on random linear regression tasks. *Remarkably, a Memformer utilizing a shared set of learned parameters is able to process batches of in-context data samples and perform competitively with, and in some cases even outperform, the CGD (and NAG) algorithm that is individually optimized for and run separately on each data sample in the test batch.*

This finding is particularly **surprising and significant** because CGD tailors its optimization individually for each data sample, whereas the Memformer applies a general optimization strategy learned from the training data across all samples. The ability of Memformers to generalize optimization strategies across data samples using shared parameters highlights their generalization capabilities, which have not been fully recognized in prior research.

(3) **Enhanced performance through multi-headed attention with theoretical insights.** We show empirically that multi-headed attention improves Memformers' test performance and offer a heuristic explanation for why increasing attention heads enhances loss performance on test data.

*Our main objective in this paper is to investigate the potential of memory-augmented Transformers to learn advanced optimization algorithms in a general sense. We are not advocating for Transformers as replacements for established optimization methods in practical applications.* Instead, we aim to shed light on the algorithmic capabilities of Transformers, inspiring further exploration into how these architectures can learn and generalize complex algorithms. We believe our results contribute to a deeper understanding of how augmented Transformers can facilitate optimization, which may ultimately lead to the discovery of new and practical gradient-based algorithms.

## 1.1 RELATED WORK

Research on Transformers is extremely active, and we cannot hope to fully capture the breadth of the related literature. Below, we summarize the most immediately relevant topics.

**In-Context Learning.** The ability of Transformer models to perform in-context learning (ICL) has been extensively studied since its introduction by Brown (2020). Subsequent works have explored how these models adapt to new tasks without requiring parameter updates (Xie et al., 2021; Von Oswald et al., 2023b; Hahn and Goyal, 2023; Liu et al., 2021; Lu et al., 2021; Wei et al., 2022; Wu et al., 2022). This foundational research has paved the way for studies investigating how Transformers can implement specific algorithms, such as gradient-based methods.

**Gradient-Based Methods in Transformers.** Garg et al. (2022) analyze the learning of gradient descent within Transformers, particularly in the context of ICL for linear functions. Empirical studies (Garg et al., 2022; Akyürek et al., 2022; Von Oswald et al., 2023a) have shown that Transformers can learn gradient descent after being trained on random linear regression tasks. Expanding on these results, Von Oswald et al. (2023a); Ahn et al. (2024) demonstrate that Transformers can implement preconditioned gradient descent for solving linear regression problems presented in input prompts. Notably, these works—as well as ours—utilize Linear Transformers as discussed in (Schlag et al., 2021; Von Oswald et al., 2023a; Ahn et al., 2023).

**Higher-Order Optimization Methods in Transformers.** Transformers have also been shown to learn higher-order optimization techniques, such as Newton's method, expanding their capabilities beyond first-order methods (Fu et al., 2023; Giannou et al., 2024; Vladymyrov et al., 2024).

**Memory-Augmented Transformers (Memformers).** Memformers were introduced by Wu et al. (2020); Xu et al. (2021). These models retain intermediate attention values across layers through memory registers, enabling more complex computations and optimization methods to be learned. While significant progress has been made in understanding how Transformers can learn gradient descent, their potential for learning more sophisticated LFOMs remains largely unexplored. Our work addresses this gap by showing how Memformers can efficiently implement a wide range of advanced first-order and quasi-second-order optimization techniques, including CGD and momentum methods, thereby pushing the boundaries of Transformer-based architectures.

## 2 BACKGROUND AND PROBLEM SETUP

### 2.1 LINEAR TRANSFORMERS ON RANDOM LINEAR REGRESSION

We follow the setup of training Transformers on random instances of linear regression, following the prior works (Garg et al., 2022; Akyürek et al., 2022; Von Oswald et al., 2023a; Ahn et al., 2024). We largely use the notation and formal setup of (Ahn et al., 2024), which we now proceed to recall.

**Data Distribution.** Let $\mathbf{x}(i) \in \mathbb{R}^d$ represent covariates drawn independently from a distribution $\mathcal{D}_{\mathbf{X}}$, and let $\mathbf{w}^* \in \mathbb{R}^d$ be drawn from $\mathcal{D}_{\mathbf{W}}$. The matrix of covariates $\mathbf{X} \in \mathbb{R}^{(n+1) \times d}$ contains rows $\mathbf{x}(i)$. The responses are $\mathbf{y} = [\langle \mathbf{x}(1), \mathbf{w}^* \rangle, \ldots, \langle \mathbf{x}(n), \mathbf{w}^* \rangle] \in \mathbb{R}^n$. Define the input matrix $\mathbf{Z}_0$ as:

$$\mathbf{Z}_0 = \begin{bmatrix} \mathbf{x}(1) & \mathbf{x}(2) & \cdots & \mathbf{x}(n) & \mathbf{x}(n+1) \\ \mathbf{y}(1) & \mathbf{y}(2) & \cdots & \mathbf{y}(n) & 0 \end{bmatrix} \in \mathbb{R}^{(d+1) \times (n+1)}, \tag{2}$$

where the zero corresponds to the unknown response for $\mathbf{x}(n+1)$. The task is to predict $(\mathbf{w}^*)^\top \mathbf{x}(n+1)$ using $\mathbf{Z}_0$. The training data consists of pairs $(\mathbf{Z}_0, (\mathbf{w}^*)^\top \mathbf{x}(n+1))$ for $\mathbf{x}(i) \sim \mathcal{D}_{\mathbf{X}}$ and $\mathbf{w}^* \sim \mathcal{D}_{\mathbf{W}}$.

**Self-Attention Without Softmax.** We focus on the linear self-attention layer, building on (Schlag et al., 2021; Von Oswald et al., 2023a). Let $\mathbf{Z} \in \mathbb{R}^{(d+1) \times (n+1)}$ be the input matrix of $n+1$ tokens in $\mathbb{R}^{d+1}$. Standard self-attention layer is defined as

$$\text{Attn}_{\text{smax}}(\mathbf{Z}) := W_v \mathbf{Z} M \cdot \text{smax}(\mathbf{Z}^\top W_k^\top W_q \mathbf{Z}), \tag{3}$$

where $W_v, W_k, W_q \in \mathbb{R}^{(d+1) \times (d+1)}$ are weight matrices, and $\text{smax}(\cdot)$ denotes the column-wise softmax. The masking matrix $M$ ensures that the label for $\mathbf{x}(n+1)$ is excluded is given by

$$M = \begin{bmatrix} \mathbf{I}_n & 0 \\ 0 & 0 \end{bmatrix} \in \mathbb{R}^{(n+1) \times (n+1)}. \tag{4}$$

Omitting softmax, the attention mechanism becomes

$$\text{Attn}_{P,Q}(\mathbf{Z}) := P\mathbf{Z}M(\mathbf{Z}^\top Q\mathbf{Z}), \tag{5}$$

where $P = W_v$ and $Q = W_k^\top W_q$. This simplified form, as shown in Ahn et al. (2024), can implement preconditioned gradient descent, and it is the one we also use.

**Transformer Architecture.** As in the related work, we also simplify the Transformer to consider only attention layers, using $L$ layers of linear self-attention with a residual connection. Therefore, for each layer $\ell$, the output is updated as

$$\mathbf{Z}_{\ell+1} = \mathbf{Z}_\ell + \frac{1}{n}\text{Attn}_{P_\ell,Q_\ell}(\mathbf{Z}_\ell), \quad \ell = 0, 1, \dots, L-1. \tag{6}$$

Using updates (6), with the input $\mathbf{Z}_0$, the final transformer output is

$$\text{TF}_L(\mathbf{Z}_0; \{P_\ell, Q_\ell\}_{\ell=0}^{L-1}) = -[\mathbf{Z}_L]_{(d+1),(n+1)}. \tag{7}$$

The set of parameters $\{P_\ell, Q_\ell\}_{\ell=0}^{L-1}$ is then learned by minimizing the following training objective:

$$f\left(\{P_\ell, Q_\ell\}_{\ell=0}^{L-1}\right) = \mathbb{E}_{(\mathbf{Z}_0, \mathbf{w}^*)}\left[\left(\text{TF}_L(\mathbf{Z}_0) + (\mathbf{w}^*)^\top \mathbf{x}(n+1)\right)^2\right]. \tag{8}$$

Here, the scaling factor $\frac{1}{n}$ is used only for ease of notation and does not influence the expressive power of the Transformer.

We will utilize the following lemma from Ahn et al. (2024), which demonstrates that multi-layer Transformers simulate preconditioned gradient descent under suitable parameterization. We have provided the full proof of this Lemma 1 in the Appendix for completeness.

$$P_\ell = \begin{bmatrix} \mathbf{B}_\ell = 0_{d\times d} & 0 \\ 0 & 1 \end{bmatrix}, \quad Q_\ell = -\begin{bmatrix} \mathbf{A}_\ell & 0 \\ 0 & 0 \end{bmatrix}, \quad \mathbf{A}_\ell, \mathbf{B}_\ell \in \mathbb{R}^{d\times d}. \tag{9}$$

**Lemma 1** (Lemma 1, Ahn et al. (2024)). *Consider an $L$-layer linear transformer parameterized by $\mathbf{A}_0, \dots, \mathbf{A}_{L-1}$, as in (9). Let $y_\ell^{(n+1)}$ be the $(d+1, n+1)$-th entry of the $\ell$-th layer output, i.e., $y_\ell^{(n+1)} = [\mathbf{Z}_\ell]_{(d+1),(n+1)}$ for $\ell = 1, \dots, L$.*

$$y_\ell^{(n+1)} = -\langle \mathbf{x}^{(n+1)}, \mathbf{w}_\ell^{\text{gd}} \rangle, \tag{10}$$

*where the sequence $\{\mathbf{w}_\ell^{\text{gd}}\}$ is defined as $\mathbf{w}_0^{\text{gd}} = 0$ and for $\ell = 1, \dots, L-1$:*

$$\mathbf{w}_{\ell+1}^{\text{gd}} = \mathbf{w}_\ell^{\text{gd}} - \mathbf{A}_\ell \nabla R_{\mathbf{w}^*}(\mathbf{w}_\ell^{\text{gd}}), \tag{11}$$

*with the empirical least-squares loss (with $\mathbf{X} := [\mathbf{x}^{(1)}, \dots, \mathbf{x}^{(n)}] \in \mathbb{R}^{d\times n}$):*

$$R_{\mathbf{w}^*}(\mathbf{w}) := \frac{1}{2n}\|\mathbf{X}^\top \mathbf{w} - \mathbf{X}^\top \mathbf{w}^*\|^2 = \frac{1}{2n}(\mathbf{w} - \mathbf{w}^*)^\top \mathbf{X}\mathbf{X}^\top(\mathbf{w} - \mathbf{w}^*). \tag{12}$$

## 2.2 LINEAR FIRST-ORDER METHODS

Linear First-Order Methods (LFOMs) (Goh, 2017) are a class of optimization algorithms that lineary combine past gradients for minimizing smooth objective functions. They iteratively update a parameter vector $\mathbf{w}$ using the gradient of the objective function. The general update rule is

$$\mathbf{w}^{k+1} = \mathbf{w}^k + \alpha_k \mathbf{d}^k, \tag{13}$$

where $\alpha_k$ is the step size and $\mathbf{d}^k$ is the update direction, typically related to the gradient $\nabla f(\mathbf{w}^k)$. Algorithms within this family differ in how they compute $\mathbf{d}^k$ and choose $\alpha_k$.

LFOMs can be expressed in a cumulative form. For gradient descent, unrolling (13) we get

$$\mathbf{w}^{k+1} = \mathbf{w}^0 - \alpha \sum_{i=0}^{k} \nabla f(\mathbf{w}^i), \tag{14}$$

while common momentum methods need an additional term incorporating past gradients, yielding

$$\mathbf{w}^{k+1} = \mathbf{w}^0 + \sum_{i=0}^{k} \gamma_i^k \nabla f(\mathbf{w}^i), \tag{15}$$

where the coefficients $\gamma_i^k$ weight previous gradients. More advanced methods, or general LFOMs, use diagonal matrices $\Gamma_i^k$ to coordinate-wise scale each gradient component, i.e.,

$$\mathbf{w}^{k+1} = \mathbf{w}^0 + \sum_{i=0}^{k} \Gamma_i^k \nabla f(\mathbf{w}^i). \tag{16}$$

**Momentum Methods and Conjugate Gradient Descent (CGD)**   Momentum methods accelerate convergence by incorporating a momentum term, modifying the gradient to account for past updates and achieving faster convergence in relevant directions. Conjugate Gradient Descent (CGD), on the other hand, is a first-order method optimized for quadratic minimization, serving as a benchmark for large-scale, sparse linear systems. After an initial steepest descent, CGD generates directions conjugate to previous ones, leading to faster convergence than standard gradient descent. Both are core methods within the LFOM class, summarized below:

**Momentum Methods**

1: Initialize $\mathbf{w}_0, \mathbf{v}_0 = 0$
2: **for** $n = 1, 2, \dots$ **do**
3:     Compute the gradient:
$$\nabla f(\mathbf{w}_n)$$
4:     Update the velocity:
$$\mathbf{v}_n = \beta \mathbf{v}_{n-1} - \eta \nabla f(\mathbf{w}_n)$$
5:     Update the iterate:
$$\mathbf{w}_{n+1} = \mathbf{w}_n + \mathbf{v}_n$$
6: **end for**
7: $\beta$: Momentum coefficient (controls the influence of past gradients)
8: $\eta$: Learning rate (scales the gradient step size)

**Conjugate Gradient Descent (CGD)**

1: Initialize $\mathbf{w}_0, \mathbf{s}_0 = -\nabla f(\mathbf{w}_0)$
2: **for** $n = 1, 2, \dots$ **do**
3:     Compute the steepest descent direction:
$$\Delta \mathbf{w}_n = -\nabla f(\mathbf{w}_n)$$
4:     Compute the conjugacy coefficient:
$$\gamma_n = \frac{\|\nabla f(\mathbf{w}_n)\|^2}{\|\nabla f(\mathbf{w}_{n-1})\|^2}$$
5:     Update the search direction:
$$\mathbf{s}_n = \Delta \mathbf{w}_n + \gamma_n \mathbf{s}_{n-1}$$
6:     Perform a line search:
$$\alpha_n = \arg\min_{\alpha} f(\mathbf{w}_n + \alpha \mathbf{s}_n)$$
7:     Update the iterate:
$$\mathbf{w}_{n+1} = \mathbf{w}_n + \alpha_n \mathbf{s}_n$$
8: **end for**

Momentum methods provide fast convergence by accumulating gradient history and are widely used in modern optimization. CGD converges in at most $N$ iterations for quadratic functions, where $N$ is the number of variables, and is effective for ill-conditioned problems.

## 3   MEMFORMERS CAN IMPLEMENT LFOMS IN-CONTEXT

Memformers can "learn" LFOMs in the specific sense discussed earlier in Section 1. Each layer $\ell$ of the Memformer has learnable parameters such as $\mathbf{A}_\ell, \mathbf{B}_\ell$ (9), and $\alpha_\ell, \gamma_\ell$ (18) or $\Gamma_\ell$ (20).

Theoretically, in Propositions 1 and 2 below, we show that in their forward pass, under certain parameter configurations, Memformers can implement exact CGD and LFOM iterations. This is indicative of the algorithmic capacities of these architectures. **In experiments, using a small number of learned parameters that are shared across a batch of in-context test data samples, the Memformer can then perform "CGD-like" (3.1) or "LFOM-like" (3.2) iterations that are competitive with, and in some cases even outperform, CGD.**

As noted in (Ahn et al., 2024, Subsection C.1), the term $\text{Attn}_{P_\ell, Q_\ell}(\mathbf{Z}_\ell)$ in the update for $\mathbf{Z}_{\ell+1}$ (6) corresponds to the preconditioned gradient $\mathbf{A}_\ell \nabla R_{\mathbf{w}^*}(\mathbf{w}_\ell^{\text{gd}})$ of the in-context loss (12) in the update for $\mathbf{w}_{\ell+1}^{\text{gd}}$.

We will henceforth call the class of algorithms that the following architecture (18) can implement as **"CGD-like"**, and the class of algorithms that architecture (20) can implement as **"LFOM-like"**.

### 3.1   DYNAMIC MEMORY FOR CGD-LIKE ALGORITHMS

**Proposition 1.** *A memory-augmented Transformer can implement Conjugate Gradient Descent (CGD) in its forward pass through a dynamic memory mechanism that recursively refines search*

*directions, where the update rules are:*

$$\mathbf{R}_\ell = \text{Attn}_{P_\ell, Q_\ell}(\mathbf{Z}_\ell) + \gamma_\ell \mathbf{R}_{\ell-1}, \quad (17)$$

$$\mathbf{Z}_{\ell+1} = \mathbf{Z}_\ell + \alpha_\ell \frac{1}{n} \mathbf{R}_\ell, \quad (18)$$

*where $\gamma_\ell$ and $\alpha_\ell$ control the influence of past updates and the step size, respectively.*

*Proof Sketch.* Here $\mathbf{R}_\ell$ denotes the state of a *single* memory register $\mathbf{R}$ at different layers $\ell$ during a forward pass. CGD refines search directions using current gradients and previous directions. The Transformer simulates this by using $\text{Attn}_{P_\ell, Q_\ell}(\mathbf{Z}_\ell)$ as the current update, analogous to the gradient in CGD, and $\gamma_\ell \mathbf{R}_{\ell-1}$ to refine the previous search direction, corresponding to the recursive update of $\mathbf{s}_n$ in CGD.

The recursive update for $\mathbf{R}_\ell$ thus mimics $\mathbf{s}_n$, the search direction in CGD. The update for $\mathbf{Z}_{\ell+1}$ uses $\mathbf{R}_\ell$, scaled by $\alpha_\ell$, similar to how CGD iterates are updated using $\mathbf{s}_n$. With $\mathbf{A}_\ell = \mathbf{I}$, this process matches CGD applied to the loss $R_{\mathbf{w}^*}(\mathbf{w})$ (12), using both current and previous gradients to refine the search direction. (**A full proof of Proposition 1 is provided in Appendix A.**) □

### 3.2 IMPLEMENTING $k$ STEPS OF LFOM WITH MEMORY REGISTERS

We extend our analysis to show how Transformers can simulate $k$ steps of Linear First-Order Methods (LFOMs). This is achieved by maintaining a memory register at each layer, which stores accumulated updates from previous layers, simulating iterative optimization.

**Proposition 2.** *A memory-augmented Transformer can implement $k$ steps of LFOM in its forward pass by maintaining memory registers across layers, where the update rules are:*

$$\mathbf{R}_\ell = \text{Attn}_{P_\ell, Q_\ell}(\mathbf{Z}_\ell), \quad (19)$$

$$\mathbf{Z}_{\ell+1} = \mathbf{Z}_\ell + \frac{1}{n} \sum_{j=0}^{\ell} \Gamma_j^\ell \odot \mathbf{R}_j, \quad (20)$$

*where $\Gamma_j^\ell$ governs the contribution of previous layers, and $\odot$ is the Hadamard product for scaling.*

*Proof Sketch.* Here each $\mathbf{R}_\ell$ denotes a *separate* memory register for each layer $\ell$. Memformers with this architecture simulate iterative optimization by refreshing the memory register $\mathbf{R}_\ell$ at each layer with $\text{Attn}_{P_\ell, Q_\ell}(\mathbf{Z}_\ell)$, capturing the current update. The cumulative update to $\mathbf{Z}_{\ell+1}$ incorporates past layers through a weighted sum of previous memory registers $\mathbf{R}_j$, with weights $\Gamma_j^\ell \in \mathbb{R}^{(d+1) \times (n+1)}$, mimicking LFOM's cumulative iterative process. We will henceforth refer to this architecture (20) as "**LFOM Memformer**".

The Hadamard product $\odot$ modulates the influence of $\mathbf{R}_j$, analogous to gradient preconditioning. This setup subsumes the case of diagonal preconditioners $\Lambda_i^k$ acting on gradients $\nabla R_{\mathbf{w}^*}(\mathbf{w}_i^{\text{gd}})$, which in the general form looks like:

$$\mathbf{w}_{k+1}^{\text{gd}} = \mathbf{w}_0 + \sum_{i=0}^{k} \Lambda_i^k \nabla R_{\mathbf{w}^*}(\mathbf{w}_i^{\text{gd}}). \quad (21)$$

The matrices $\Gamma_j^\ell \in \mathbb{R}^{(d+1) \times (n+1)}$ and $\Lambda_i^k \in \mathbb{R}^{d \times d}$ serve similar roles, but their dimensions differ. We expect this Hadamard product memory architecture to be able to perform richer algorithms than LFOMs, though a formal characterization of its full potential remains to be done.

The full proof follows from the cumulative memory structure and the connection between attention and preconditioned gradients, as discussed in the proof steps of Lemma 1. (**A full proof of Proposition 2 is provided in Appendix A.**) □

**Remark.** The update (20) could be interpreted as a type of *gated memory*, related to gating in LSTMs and GRUs that also use the Hadamard product to modulate information flow through gates. This similarity suggests that principles from these architectures could help refine memory mechanisms in Transformers, potentially enhancing their ability to handle long-term dependencies in optimization tasks. However, further exploration is needed to fully understand this relationship.

### 3.3 Experimental Results: Memformer Performance vs. CGD

In this section, we present our empirical results for Memformers "learning" conjugate gradient descent (CGD), general linear first-order methods (LFOMs), and general LFOMs with $GD^{++}$. The method $GD^{++}$ is a quasi-Newton method where the inverse Hessian in Newton's method is approximated by a truncated Neumann series; for more details on $GD^{++}$, refer to Section A.10 of Von Oswald et al. (2023a).

We consider the in-context loss function (12) for linear regression. The input dimension is set to $d = 5$, and the number of training observations in the prompt is $n = 20$. Both the inputs $\mathbf{x}^{(i)}$ and the target weight vector $\mathbf{w}^*$ are sampled from Gaussian distributions: $\mathbf{x}^{(i)} \sim \mathcal{N}(0, \boldsymbol{\Sigma})$ and $\mathbf{w}^* \sim \mathcal{N}(0, \boldsymbol{\Sigma}^{-1})$, where $\boldsymbol{\Sigma} = \mathbf{U}^\top \mathbf{D} \mathbf{U}$. Here, $\mathbf{U}$ is a uniformly random orthogonal matrix, and $\mathbf{D}$ is a fixed diagonal matrix with entries $\mathrm{diag}(1, 1, 1/2, 1/4, 1)$.

We optimize the function $f$ (8) for a three-layer linear transformer using the ADAM optimizer. The matrices $\mathbf{A}_0$, $\mathbf{A}_1$, and $\mathbf{A}_2$ (as in (9)) are initialized with independent and identically distributed (i.i.d.) Gaussian entries. Each gradient step is computed using a batch of size 1000, and we resample the batch every 100 steps. We clip the gradient of each matrix to have a maximum norm of 0.01. All plots are averaged over five runs, each with a different randomly sampled $\mathbf{U}$ (and thus different $\boldsymbol{\Sigma}$).

Figure 1 illustrates the implementation of a CGD-like algorithm under the architecture given by (18). In Figure 1a, the line-search parameters $\alpha_\ell$ and deflection parameters $\gamma_\ell$ for each layer $\ell$ are obtained by training using ADAM. By "CGD-like," we mean that upon training the Memformer using ADAM, the Memformer layers learn general parameters $\alpha_\ell$ and $\gamma_\ell$ which, while they may not match the exact CGD parameters for individual observations, perform well enough on each observation to be comparable to, if not competitive with, CGD. We further explain the important issue of learning general parameters in Section 4.

Figure 1b presents the same experiment as Figure 1a, using the architecture in (18), but with the parameters $\mathbf{A}_\ell$ for each layer not restricted to scalars. Thus, past gradients are accounted for, similar to CGD, but with preconditioners $\mathbf{A}_\ell$. This is therefore not a "CGD-like" algorithm. We aim to demonstrate that once we allow preconditioned gradients, a Memformer implements a certain "LFOM-like" algorithm that distinctly outperforms CGD.

Figure 2 presents the performance of LFOM Memformer under the architecture in (20), where the matrix parameters $\Gamma_j$ for each layer $j$ are obtained by training using ADAM. In our experiments, we consider the special case of $\Gamma_j^\ell = \Gamma_j \ \forall \ell$, which is more natural, if we consider that each layer $j$ of the Memformer has an associated $\Gamma_j$. Figure 2a shows the results on non-isotropic data, and Figure 2b shows the results on isotropic data. Note that this algorithm is quite similar in nature to the previous case in Figure 1b. Here, the $\Gamma_j$'s essentially act as preconditioners of the gradients computed in each layer. Consequently, the graphs of Figures 1b and 2a are nearly identical. In the isotropic data experiment (Figure 2b), we observe that the Memformer does not perform better than a linear transformer. In quadratics with isotropic data, there is no significant variation in curvature across directions; thus, incorporating past gradients via momentum offers little advantage. Momentum is more beneficial in cases with non-isotropic data.

Figure 3 presents LFOM Memformer with $GD^{++}$ under the architecture in (20), where the $\mathbf{B}_\ell$ blocks in the $P_\ell$ matrices for each layer $\ell$ (9) are allowed to be non-zero. Once again, the matrix parameters for each layer $\ell$ are obtained by training using ADAM. In this case, the $\mathbf{B}_\ell$ matrices resemble a heavily truncated Neumann series of the inverse $\mathbf{X}\mathbf{X}^\top$ (Hessian of (12)), resulting in a quasi-Newton method. The experiments are conducted on both non-isotropic data (Figure 3a) and isotropic data (Figure 3b).

## 4 Experiments: Influence of Batch Size on Performance

We emphasize here that the results presented in Section 3.3 compare the performance of Transformers and Memformers (which learn shared generic parameters upon training) against CGD that runs on fresh observations of batch size $B = 1000$, independently resampled from the same distribution. But unlike CGD that computes specific parameters for each observation, the Transformer and Memformer models learn shared parameters $P_\ell, Q_\ell$ (and $\alpha_\ell, \gamma_\ell$, or $\Gamma_\ell$) for each layer $\ell$, and these parameters are applied uniformly across all 1000 observations in the batch. In contrast, CGD is executed individually on each of the 1000 observations in the batch, and the average log-loss versus layers is plotted.

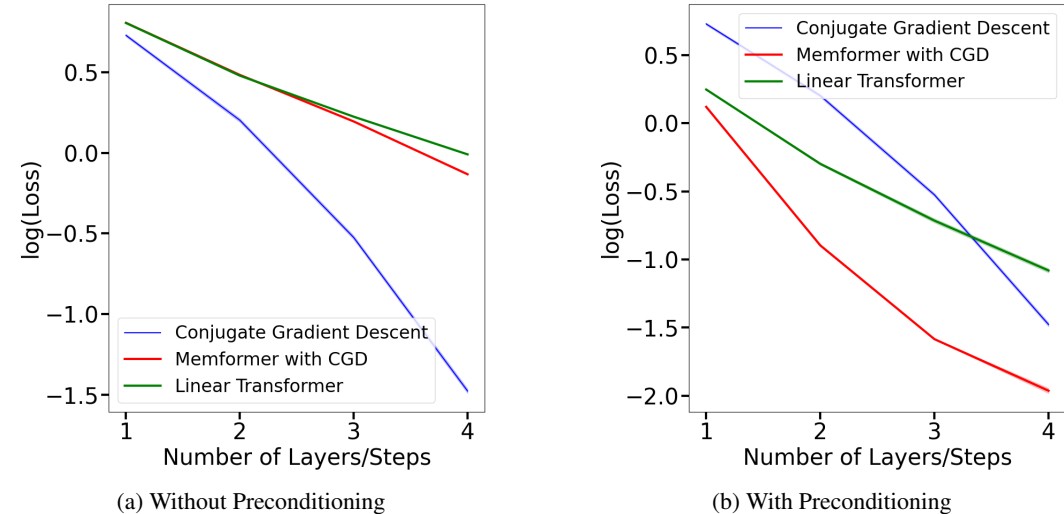

(a) Without Preconditioning        (b) With Preconditioning

Figure 1: Comparison of Linear Transformer and CGD Memformer (18) with general CGD-like parameters to actual CGD running separately on each test observation.

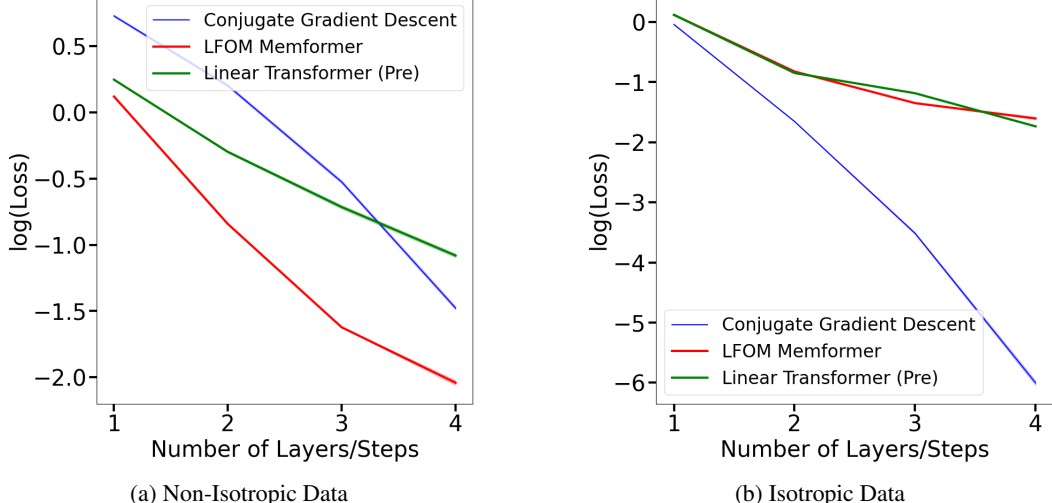

(a) Non-Isotropic Data        (b) Isotropic Data

Figure 2: LFOM Memformer (20) performance on non-isotropic vs. isotropic test data (Pre = with non-trivial preconditioners). Test data is independently sampled from the same distribution as the training data.

The strength of LFOM Memformers (20) (with matrices $\Gamma_\ell$ restricted to scalar multiples of the identity) becomes even more pronounced when tested on training data with small batch sizes, such as $B = 1$ and $B = 10$. In these scenarios, the Memformers learn parameters that significantly outperform CGD running in parallel on each of the observations in those small batches. Figure 4 demonstrates this comparison. We further provide an experimental comparison of LFOM Memformer performance vs. Nesterov Accelerated Gradient Method and Momentum GD in the Appendix.

## 5   EXPERIMENTS: IMPACT OF USING MULTI-HEADED ATTENTION

Our experiments show that increasing the number of attention heads improves test loss performance. Multi-head attention enables Transformers to learn diverse preconditioning matrices, better adapting to varying data covariance structures. In our architecture (17), attention values from each head are summed into the memory register $\mathbf{R}_\ell$ at each layer. Heuristically, each head captures different aspects of the data, estimating gradients from multiple perspectives. This ensemble-like behavior reduces variance in gradient updates by averaging out individual noise and biases, leading to faster

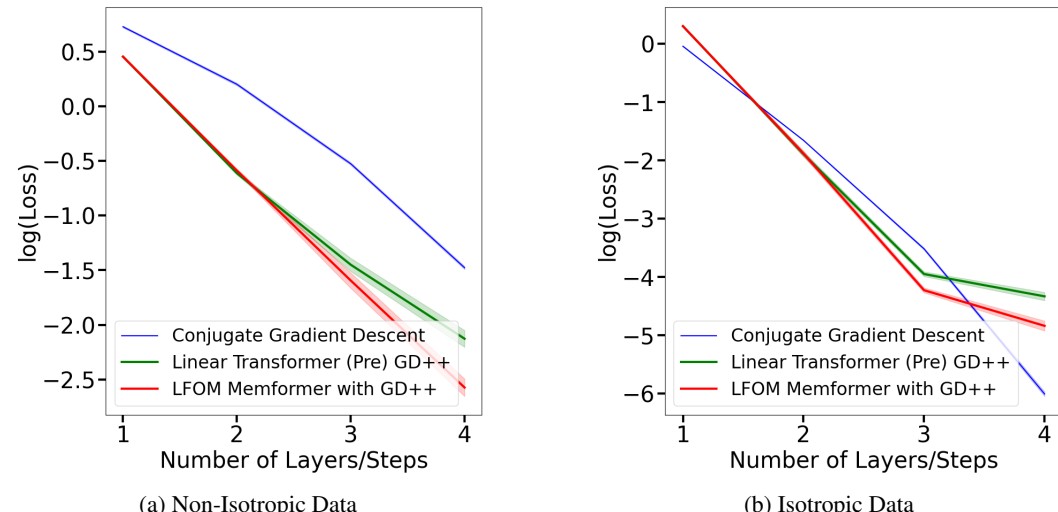

(a) Non-Isotropic Data      (b) Isotropic Data

Figure 3: LFOM Memformer [20] GD++ performance on non-isotropic vs. isotropic test data (Pre = with non-trivial preconditioners). Test data is independently sampled from the same distribution as the training data.

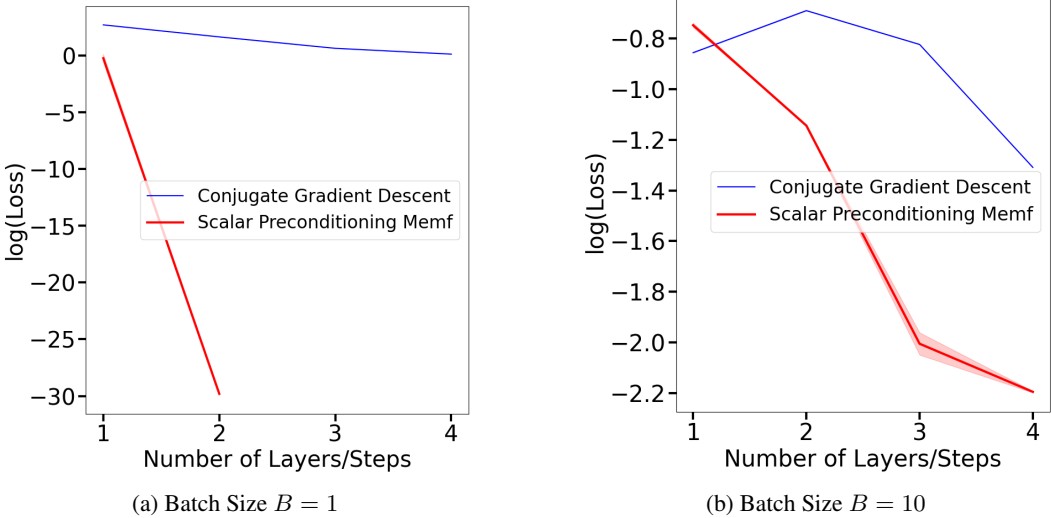

(a) Batch Size $B = 1$      (b) Batch Size $B = 10$

Figure 4: LFOM Memformer [20] with scalar preconditioners $\Gamma_\ell$ vs. CGD performance on small batch training data ($B = 1$ and $B = 10$). The Memformer demonstrates superior performance on the training data.

convergence and more stable optimization. Acting as implicit regularization, it prevents overfitting and enhances generalization on test data. This phenomenon is also supported by recent studies. Chen et al. (2024) showed that multi-head attention is essential for effective context preprocessing in sparse linear regression, aligning with our findings. Similarly, Cui et al. (2024) provided theoretical and empirical evidence that multi-head attention outperforms single-head attention in in-context learning.

Figure 5 compares models with 1-head and 5-head attention, illustrating the benefits of multiple heads on convergence speed and test loss performance.

## 6 DISCUSSION AND FUTURE WORK

This work demonstrates the capability of memory-augmented Transformers (Memformers) to implement a broad range of first-order optimization methods, opening several research directions. We briefly comment on some of these aspects below.

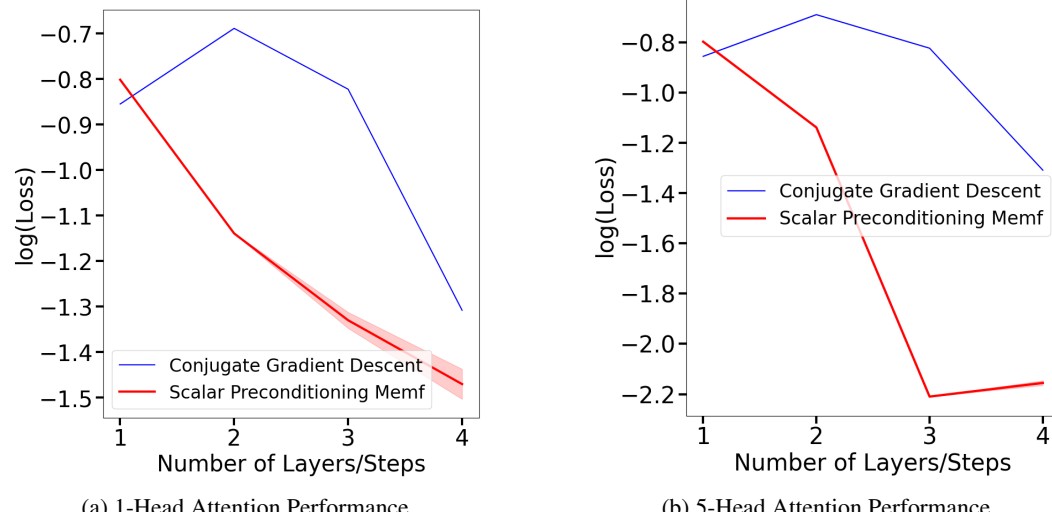

(a) 1-Head Attention Performance  (b) 5-Head Attention Performance

Figure 5: Comparison of LFOM Memformer (20) (with scalar preconditioners $\Gamma_\ell$) performance using 1-head and 5-head attention, relative to CGD.

(i) **Architectural Flexibility**: Small modifications, such as (gated) memory registers, significantly enhance Transformers' ability to learn and implement diverse optimization algorithms. Future research could explore further architectural innovations to unlock even greater capabilities.

(ii) **General Function Classes**: While our approach successfully makes Transformers implement LFOMs on quadratic functions, future work should extend this to more general objective functions. Doing so may require novel training strategies, and possibly architectural adjustments to handle non-quadratic functions. The role of nonlinear attention and the MLP component of Transformers may also prove to be useful here.

(iii) **Efficiency vs. Generalization**: Attention-based methods require more computation than directly implementing conjugate gradient descent or momentum GD. However, Transformers excel in learning general parameters, enabling LFOMs to generalize across new data without needing per-instance optimization. Exploring practical use of such "learned optimizers" to either warmstart a solver, or to potentially even bypass it, is a tantalizing research topic.

(iv) **Theoretical Foundations and Convergence Analysis**: Strengthening the theoretical basis of Transformers' optimization capabilities, including convergence analysis and their alignment with classical optimization theory, is another important direction for future research.

(v) **Meta-learning and Transfer Learning**: The ability of Transformers to learn and generalize optimization algorithms offers exciting potential for meta-learning and transfer learning, providing new opportunities in areas where traditional optimization methods fall short.

## 6.1 LIMITATIONS

We briefly remark on some limitations of our current framework. For instance, while Memformers are quite versatile, our experiments (Figures 1, 2) indicate they do not radically outperform preconditioned GD on general quadratic problems as in (12), where the preconditioner matrix $\Gamma_\ell$ (and likewise, $\mathbf{A}_\ell$) for the current layer $\ell$ is the main contributor to loss performance at each update step $\ell$ (17). On the other hand, this behavior is likely due to the task being quadratic, and a future study that tackles more general ICL formulations will likely shed light here.

Transformers can implement second-order methods like Newton's method (Fu et al., 2023; Giannou et al., 2024), which typically outperform LFOMs in convergence speed and accuracy. However, we reiterate that the main focus of our paper is to explore the space of first-order optimization algorithms that augmented Transformers can learn, as opposed to looking for "the best" algorithm.

## 7 REPRODUCIBILITY STATEMENT

We believe the following points provide a clear path for replicating our results:

- **Code Availability**: The code for our experiments, including Memformers and LFOM implementations, is available at https://anonymous.4open.science/r/ICLR-2025-Memformer_LFOM.

- **Experiment Setup**: Detailed descriptions of the training setup, model architecture, parameter initialization, and optimization methods are included in Sections 2 and 3.3.

- **Random Seeds**: Random seeds were fixed across all experiments to ensure consistency, and they are provided in the code repository for replication.

- **Hardware Requirements**: All experiments were conducted on NVIDIA T4 GPUs in Google Colab.

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

# SUPPLEMENTARY MATERIAL

## A PROOFS

### A.1 PROOF OF LEMMA 1: EQUIVALENCE TO PRECONDITIONED GRADIENT DESCENT

This proof already exists in the literature, for instance, in Subsection C.1 of Ahn et al. (2024). However, we repeat it here, to make this paper as self-contained as possible.

Consider a set of fixed samples $\mathbf{x}^{(1)}, \ldots, \mathbf{x}^{(n)}$, along with a fixed vector $\mathbf{w}^*$. Let $P = \{P_i\}_{i=0}^k$ and $Q = \{Q_i\}_{i=0}^k$ represent fixed weights, and let $\mathbf{Z}_i$ evolve as per equation (6). Define $\mathbf{X}_i$ as the first $d$ rows of $\mathbf{Z}_k$ (under equation (9), we have $\mathbf{X}_i = \mathbf{X}_0$ for all $i$), and let $\mathbf{Y}_i$ be the $(d+1)$-th row of $\mathbf{Z}_i$. Now, let $g(\mathbf{x}, \mathbf{y}, k) : \mathbb{R}^d \times \mathbb{R} \times \mathbb{Z} \to \mathbb{R}$ be a function such that for $\mathbf{x}_{n+1} = \mathbf{x}$ and $\mathbf{y}_{n+1}^{(0)} = \mathbf{y}$, the function is defined as $g(\mathbf{x}, \mathbf{y}, k) := \mathbf{y}_{n+1}^{(k)}$. It's worth noting that $\mathbf{y}_{n+1}^{(k)} = [\mathbf{Y}_k]_{n+1}$.

We can verify that, under equation (9), the update rule for $\mathbf{y}_{n+1}^{(k)}$ is given by:

$$\mathbf{Y}_{k+1} = \mathbf{Y}_k - \frac{1}{n}\mathbf{Y}_k M \mathbf{X}_0^\top A_k \mathbf{X}_0, \tag{22}$$

where $M$ is a mask matrix of the form:

$$M = \begin{bmatrix} I & 0 \\ 0 & 0 \end{bmatrix}.$$

The following points can be verified:

1. $g(\mathbf{x}, \mathbf{y}, k) = g(\mathbf{x}, 0, k) + \mathbf{y}$. To see this, note that for each $i \in \{1, \ldots, n\}$, we have:

$$\mathbf{y}_{k+1}^{(i)} = \mathbf{y}_k^{(i)} - \frac{1}{n}\sum_{j=1}^n \mathbf{x}^{(i)\top} A_k \mathbf{x}^{(j)} \mathbf{y}_k^{(j)}.$$

Thus, $\mathbf{y}_k^{(i)}$ does not depend on $\mathbf{y}_{n+1}^{(t)}$ for any $t$. For $\mathbf{y}_{n+1}^{(k)}$, the update becomes:

$$\mathbf{y}_{n+1}^{(k+1)} = \mathbf{y}_{n+1}^{(k)} - \frac{1}{n}\sum_{j=1}^n \mathbf{x}_{n+1}^\top A_k \mathbf{x}^{(j)} \mathbf{y}_k^{(j)},$$

which clearly shows that the dependence on $\mathbf{y}_{n+1}^{(k)}$ is additive. Through a simple induction, we can establish:

$$g(\mathbf{x}, \mathbf{y}, k+1) - \mathbf{y} = g(\mathbf{x}, \mathbf{y}, k) - \mathbf{y}.$$

2. The function $g(\mathbf{x}, 0, k)$ is linear in $\mathbf{x}$. To see this, note that for $j \neq n+1$, $\mathbf{y}_j^{(k)}$ does not depend on $\mathbf{x}_{n+1}^{(t)}$ for any $t$, $j$, or $k$. Therefore, the update for $\mathbf{y}_{n+1}^{(k+1)}$ depends linearly on $\mathbf{x}_{n+1}$ and $\mathbf{y}_{n+1}^{(k)}$. Since $\mathbf{y}_{n+1}^{(0)} = 0$ is linear in $\mathbf{x}$, we conclude by induction that the result holds.

Considering these points, we can confirm that for each $k$, there exists a vector $\theta_k \in \mathbb{R}^d$ such that:

$$g(\mathbf{x}, \mathbf{y}, k) = g(\mathbf{x}, 0, k) + \mathbf{y} = \langle \theta_k, \mathbf{x} \rangle + \mathbf{y},$$

for all $\mathbf{x}$ and $\mathbf{y}$. It follows that $g(\mathbf{x}, \mathbf{y}, 0) = \mathbf{y}$, so that $\langle \theta_0, \mathbf{x} \rangle = g(\mathbf{x}, \mathbf{y}, 0) - \mathbf{y} = 0$, implying $\theta_0 = 0$.

We now focus on the third key fact: for each $i$, we have:

$$g(\mathbf{x}^{(i)}, \mathbf{y}^{(i)}, k) = \mathbf{y}_k^{(i)} = \langle \theta_k, \mathbf{x}^{(i)} \rangle + \mathbf{y}^{(i)}.$$

To prove this, let $\mathbf{x}_{n+1} := \mathbf{x}^{(i)}$ for some $i \in \{1, \ldots, n\}$. Then:

$$\mathbf{y}_{k+1}^{(i)} = \mathbf{y}_k^{(i)} - \frac{1}{n}\sum_{j=1}^n \mathbf{x}^{(i)\top} A_k \mathbf{x}^{(j)} \mathbf{y}_k^{(j)},$$

$$\mathbf{y}_{n+1}^{(k+1)} = \mathbf{y}_{n+1}^{(k)} - \frac{1}{n}\sum_{j=1}^{n}\mathbf{x}_{n+1}^{\top}A_k\mathbf{x}^{(j)}\mathbf{y}_k^{(j)},$$

therefore, $\mathbf{y}_{k+1}^{(i)} = \mathbf{y}_{n+1}^{(k+1)}$ when $\mathbf{y}_k^{(i)} = \mathbf{y}_{n+1}^{(k)}$. This completes the induction, given that $\mathbf{y}_0^{(i)} = \mathbf{y}_{n+1}^{(0)}$ by definition.

Let $\bar{\mathbf{X}} \in \mathbb{R}^{d\times n}$ be the matrix whose columns are $\mathbf{x}^{(1)}, \ldots, \mathbf{x}^{(n)}$, excluding $\mathbf{x}_{n+1}$, and let $\bar{\mathbf{Y}}_k \in \mathbb{R}^{1\times n}$ be the vector of $\mathbf{y}_k^{(1)}, \ldots, \mathbf{y}_k^{(n)}$. It follows that:

$$\bar{\mathbf{Y}}_k = \bar{\mathbf{Y}}_0 + \theta_k^{\top}\bar{\mathbf{X}}.$$

Using this, the update formula for $\mathbf{y}_{n+1}^{(k)}$ becomes:

$$\mathbf{y}_{n+1}^{(k+1)} = \mathbf{y}_{n+1}^{(k)} - \frac{1}{n}\langle A_k\bar{\mathbf{X}}^{\top}\bar{\mathbf{Y}}_k, \mathbf{x}_{n+1}\rangle, \tag{23}$$

leading to the update:

$$\langle\theta_{k+1}, \mathbf{x}_{n+1}\rangle = \langle\theta_k, \mathbf{x}_{n+1}\rangle - \frac{1}{n}\langle A_k\bar{\mathbf{X}}(\bar{\mathbf{X}}^{\top}\theta_k + \bar{\mathbf{Y}}_0), \mathbf{x}_{n+1}\rangle. \tag{24}$$

Since $\mathbf{x}_{n+1}$ is arbitrary, we derive the general update formula:

$$\theta_{k+1} = \theta_k - \frac{1}{n}A_k\bar{\mathbf{X}}\bar{\mathbf{X}}^{\top}(\theta_k + \mathbf{w}^*). \tag{25}$$

Treating $A_k$ as a preconditioner, and letting $f(\theta) := \frac{1}{2n}(\theta + \mathbf{w}^*)^{\top}\bar{\mathbf{X}}\bar{\mathbf{X}}^{\top}(\theta + \mathbf{w}^*)$, we can express the update as:

$$\theta_{k+1} = \theta_k - \frac{1}{n}A_k\nabla f(\theta). \tag{26}$$

Finally, let $\mathbf{w}_k^{\mathrm{gd}} := -\theta_k$. We can verify that $f(-\mathbf{w}) = R_{\mathbf{w}^*}(\mathbf{w})$, implying that:

$$\mathbf{w}_{k+1}^{\mathrm{gd}} = \mathbf{w}_k^{\mathrm{gd}} - \frac{1}{n}A_k\nabla R_{\mathbf{w}^*}(\mathbf{w}_k^{\mathrm{gd}}). \tag{27}$$

We also confirm that for any $\mathbf{x}_{n+1}$, the prediction of $\mathbf{y}_{n+1}^{(k)}$ is:

$$g(\mathbf{x}_{n+1}, \mathbf{y}_{n+1}, k) = \mathbf{y}_{n+1} - \langle\theta, \mathbf{x}_{n+1}\rangle = \mathbf{y}_{n+1} + \langle\mathbf{w}_k^{\mathrm{gd}}, \mathbf{x}_{n+1}\rangle.$$

This concludes the proof. We have simply followed the update rule (6) to its logical conclusion.

## A.2 FULL PROOF OF PROPOSITION 1

*A memory-augmented Transformer can implement Conjugate Gradient Descent (CGD) through a dynamic memory mechanism that recursively refines search directions, where the update rules are:*

$$\mathbf{R}_\ell = \mathrm{Attn}_{P_\ell, Q_\ell}(\mathbf{Z}_\ell) + \gamma_\ell\mathbf{R}_{\ell-1}, \tag{28}$$

$$\mathbf{Z}_{\ell+1} = \mathbf{Z}_\ell + \alpha_\ell\frac{1}{n}\mathbf{R}_\ell, \tag{29}$$

*where $\gamma_\ell$ and $\alpha_\ell$ control past update influence and step size.*

### PROOF

Our goal is to demonstrate that, under appropriate parameter configurations, the memory-augmented Transformer updates given by equations (28) and (29) correspond precisely to the Conjugate Gradient Descent (CGD) algorithm when applied to the quadratic loss function:

$$R_{\mathbf{w}^*}(\mathbf{w}) = \frac{1}{2n}(\mathbf{w} - \mathbf{w}^*)^{\top}\mathbf{X}\mathbf{X}^{\top}(\mathbf{w} - \mathbf{w}^*). \tag{30}$$

We will establish a mapping between the Transformer's operations and the steps of the CGD algorithm, demonstrating that the Transformer can implement CGD under certain parameter settings.

### CGD ALGORITHM FOR QUADRATIC FUNCTIONS

For minimizing a quadratic function, the CGD algorithm proceeds as follows:

---

**Algorithm**. Conjugate Gradient Descent (CGD)

---

Initialize $\mathbf{w}_0$ and $\mathbf{r}_0 = -\nabla f(\mathbf{w}_0)$, $\mathbf{s}_0 = \mathbf{r}_0$

$\mathbf{w}_1 = \mathbf{w}_0 + \mathbf{r}_0$

**for** $n = 1, 2, \ldots$ **do**

 Compute the residual: $\mathbf{r}_n = -\nabla f(\mathbf{w}_n)$

 Compute the conjugacy coefficient:

$$\gamma_n = \frac{\mathbf{r}_n^\top \mathbf{r}_n}{\mathbf{r}_{n-1}^\top \mathbf{r}_{n-1}}$$

 Update the search direction:

$$\mathbf{s}_n = \mathbf{r}_n + \gamma_n \mathbf{s}_{n-1}$$

 Compute the step size:

$$\alpha_n = \frac{\mathbf{r}_n^\top \mathbf{r}_n}{\mathbf{s}_n^\top \mathbf{H} \mathbf{s}_n}$$

 Update the parameters:

$$\mathbf{w}_{n+1} = \mathbf{w}_n + \alpha_n \mathbf{s}_n$$

**end for**

---

MAPPING CGD UPDATES TO TRANSFORMER UPDATES

We first recall that in the proof of Lemma 1 (A.1), the $\mathbf{w}_{k+1}^{\mathrm{gd}}$ update rule

$$\mathbf{w}_{k+1}^{\mathrm{gd}} = \mathbf{w}_k^{\mathrm{gd}} - \frac{1}{n} A_k \nabla R_{\mathbf{w}^*}(\mathbf{w}_k^{\mathrm{gd}}), \tag{31}$$

is a direct downstream consequence of the $\mathbf{Z}_{\ell+1}$ update rule (6)

$$\mathbf{Z}_{\ell+1} = \mathbf{Z}_\ell + \frac{1}{n}\mathrm{Attn}_{P_\ell, Q_\ell}(\mathbf{Z}_\ell), \quad \ell = 0, 1, \ldots, L-1, \tag{32}$$

under the parameterization given in equation (9). Thus, the $\mathrm{Attn}_{P_\ell, Q_\ell}$ term in the $\mathbf{Z}_\ell$ update equation is, in a precise sense, paralleled by the $-\frac{1}{n} A_k \nabla R_{\mathbf{w}^*}(\mathbf{w}_k^{\mathrm{gd}})$ term in the $\mathbf{w}_{k+1}^{\mathrm{gd}}$ update equation (31).

STEP 1: INITIALIZATION

- **CGD:**
$$\mathbf{w}_0 \text{ given,} \quad \mathbf{r}_0 = -\nabla f(\mathbf{w}_0), \quad \mathbf{s}_0 = \mathbf{r}_0.$$

- **Transformer:**
  - The initial state $\mathbf{Z}_0$ in (6) parallels $\mathbf{w}_0$ in (31).
  - The memory register $\mathbf{R}$ is initialized to $\mathrm{Attn}_{P_0, Q_0}(\mathbf{Z}_0)$, i.e., $\mathbf{R}_0 = \mathrm{Attn}_{P_0, Q_0}(\mathbf{Z}_0)$, corresponding to $\mathbf{s}_0 = \mathbf{r}_0$.
  - We set $\gamma_0 = 0$, consistent with CGD initialization.

STEP 2: UPDATE MEMORY REGISTER (SEARCH DIRECTION)

- **Transformer Memory Update:**
$$\mathbf{R}_\ell = \mathrm{Attn}_{P_\ell, Q_\ell}(\mathbf{Z}_\ell) + \gamma_\ell \mathbf{R}_{\ell-1}.$$

- **Correspondence with CGD:**
$$\mathbf{s}_n = \mathbf{r}_n + \gamma_n \mathbf{s}_{n-1}.$$
  Identifying $\mathbf{R}_\ell \leftrightarrow \mathbf{s}_n$, $\gamma_\ell = \gamma_n$, and $\mathbf{R}_{\ell-1} \leftrightarrow \mathbf{s}_{n-1}$, the Transformer's memory update matches CGD.

STEP 3: UPDATE PARAMETERS

- **Transformer Parameter Update:**
$$\mathbf{Z}_{\ell+1} = \mathbf{Z}_\ell + \alpha_\ell \frac{1}{n} \mathbf{R}_\ell.$$

- **Correspondence with CGD:**

$$\mathbf{w}_{n+1} = \mathbf{w}_n + \alpha_n \mathbf{s}_n.$$

  The scaling factor $\frac{1}{n}$ accounts for the gradient's scaling, consistent with the CGD update when considering the Hessian $\mathbf{H} = \frac{1}{n}\mathbf{X}\mathbf{X}^\top$.

STEP 4: CONJUGACY COEFFICIENT $\gamma_\ell$ AND STEP SIZE $\alpha_\ell$

- **CGD Computations**: Scalar values computed based on residuals and the Hessian.
- **Transformer Implementation**:
    - $\gamma_\ell$ and $\alpha_\ell$ are treated as parameters, ensuring structural correspondence.
    - The Transformer's architecture allows these as fixed or learnable parameters.

Therefore, under suitable parameter configurations, the memory-augmented Transformer can implement CGD, demonstrating the feasibility of using the Transformer's architecture to perform CGD-like updates.

### A.3 FULL PROOF OF PROPOSITION 2

*A memory-augmented Transformer can implement $k$ steps of Linear First-Order Methods (LFOMs) by maintaining memory registers across layers, where the update rules are:*

$$\mathbf{R}_\ell = \mathrm{Attn}_{P_\ell, Q_\ell}(\mathbf{Z}_\ell), \tag{33}$$

$$\mathbf{Z}_{\ell+1} = \mathbf{Z}_\ell + \frac{1}{n}\sum_{j=0}^{\ell}\Gamma_j^\ell \odot \mathbf{R}_j, \tag{34}$$

*where $\Gamma_j^\ell$ governs the contribution of previous layers, and $\odot$ is the Hadamard (element-wise) product for scaling.*

Our goal is to show that the memory-augmented Transformer with updates given by equations (33) and (34) can implement $k$ steps of an LFOM, whose general formulation is:

$$\mathbf{w}^{k+1} = \mathbf{w}^0 + \sum_{i=0}^{k}\Lambda_i^k \nabla f(\mathbf{w}^i),$$

where $\Lambda_i^k$ are diagonal matrices that scale the gradients $\nabla f(\mathbf{w}^i)$.

We will proceed by establishing a correspondence between the variables and updates in the memory-augmented Transformer and those in the LFOM, and by showing that, under appropriate parameter settings, the Transformer updates replicate the LFOM updates.

The first order of business is to realize that, in the proof of Lemma 1 (A.1), the $\mathbf{w}_{k+1}^{\mathrm{gd}}$ update rule (31) is a direct downstream consequence of the $\mathbf{Z}_{\ell+1}$ update rule (6), under the parameterization given in equation (9).

Set $\mathbf{R}_\ell = \mathrm{Attn}_{P_\ell, Q_\ell}(\mathbf{Z}_\ell)$ per (33). Then the consequence of the $\mathbf{Z}_{\ell+1} = \mathbf{Z}_\ell + \frac{1}{n}\sum_{j=0}^{\ell}\Gamma_j^\ell \odot \mathbf{R}_j$ update rule is that each $\mathrm{Attn}_{P_j, Q_j}(\mathbf{Z}_j)$ is coordinate-wise scaled by $\Gamma_j^\ell \in \mathbb{R}^{(d+1)\times(n+1)}$. But if $\mathrm{Attn}_{P_j, Q_j}(\mathbf{Z}_j)$ is coordinate-wise scaled by $\Gamma_j^\ell$, then the $\mathbf{Y}_{k+1}$ update rule in (22) now instead looks like $\mathbf{Y}_{k+1} = \mathbf{Y}_k - \frac{1}{n}\sum_{j=0}^{k}\Gamma_j^k\big|_{d+1} \odot (\mathbf{Y}_k M \mathbf{X}_0^\top A_k \mathbf{X}_0)$, where $\Gamma_j^k\big|_{d+1}$ denotes the $(d+1)$-th row of $\Gamma_j^k$. This is because, by definition, $\mathbf{Y}_i$ is the $(d+1)$-th row of $\mathbf{Z}_i$ (A.1).

From the basic $\mathbf{Y}_k$ update rule in (22), the update formula for $\mathbf{y}_{n+1}^{(k+1)}$ in (23) follows as a consequence. Except that now, this update formula will include a coordinate-wise scaling as well, which we will denote by $\Lambda_j^k \in \mathbb{R}^d$:

$$\mathbf{y}_{n+1}^{(k+1)} = \mathbf{y}_{n+1}^{(k)} - \frac{1}{n}\sum_{j=0}^{k}\langle(A_j\bar{\mathbf{X}}^\top\bar{\mathbf{Y}}_j)\odot\Lambda_j^k, \mathbf{x}_{n+1}\rangle,$$

which in turn leads to $\theta_{k+1} = \theta_k - \frac{1}{n}\sum_{j=0}^k (A_j \bar{\mathbf{X}}\bar{\mathbf{X}}^\top (\theta_j + \mathbf{w}^*)) \odot \Lambda_j^k$ in place of (25) and $\mathbf{w}_{k+1}^{\text{gd}} = \mathbf{w}_k^{\text{gd}} - \frac{1}{n}\sum_{j=0}^k A_j \nabla R_{\mathbf{w}^*}(\mathbf{w}_j^{\text{gd}}) \odot \Lambda_j^k$ in place of (26). The negative signs can, of course, be incorporated within the $\Lambda_j^k$s.

If we simply rewrite $\Lambda_j^k \in \mathbb{R}^d$ as a diagonal matrix in $\mathbb{R}^{d \times d}$, this setup then subsumes the case of diagonal preconditioners $\Lambda_j^k \in \mathbb{R}^{d \times d}$ acting on the gradients $\nabla R_{\mathbf{w}^*}(\mathbf{w}_j^{\text{gd}})$, which in the general form looks like:

$$\mathbf{w}_{k+1}^{\text{gd}} = \mathbf{w}_0 + \sum_{i=0}^k \Lambda_i^k \nabla R_{\mathbf{w}^*}(\mathbf{w}_i^{\text{gd}}). \tag{35}$$

where $\Lambda_i^k$ are diagonal matrices.

***Note.*** The memory-augmented Transformer performs exactly these updates in the special case when the preconditioners $A_j$ are scalar multiples of the identity. If the preconditioners $A_j$ are non-trivial, then this architecture performs **"LFOM-like"** algorithms that lie in a class richer than LFOMs (3.2).

# B    COMPARISON TO NESTEROV ACCELERATED GRADIENT METHOD (NAG) AND MOMENTUM GRADIENT DESCENT (MGD)

## B.1    NESTEROV ACCELERATED GRADIENT METHOD (NAG)

NAG is a commonly used optimization technique that builds on classical gradient descent by incorporating a momentum term that anticipates the next update. Specifically, the weights are updated using the following update rules:

$$\mathbf{v}_{k+1} = \mathbf{w}_k + \beta_k(\mathbf{w}_k - \mathbf{w}_{k-1})$$

$$\mathbf{w}_{k+1} = \mathbf{v}_{k+1} - \eta_k \nabla f(\mathbf{v}_{k+1})$$

Here, $\beta_k$ controls the influence of previous updates (momentum), and $\eta_k$ is the learning rate. In our experiments, we selected $\eta_k = 0.03$ and $\beta_k = 0.9$ after testing various values of these parameters on the given distribution, as in Section 3.3. These values provided the best performance. The momentum term allows NAG to "look ahead" in the optimization trajectory, which often leads to faster convergence than vanilla gradient descent.

## B.2    MOMENTUM GRADIENT DESCENT (MGD)

Momentum Gradient Descent operates similarly to NAG but without the anticipation of future steps. The algorithm updates the weights based on a momentum term that accelerates convergence in directions with consistent gradients. The update rule for MGD is given by:

$$\mathbf{v}_{k+1} = \beta_k \mathbf{v}_k - \eta_k \nabla f(\mathbf{w}_k)$$

$$\mathbf{w}_{k+1} = \mathbf{w}_k + \mathbf{v}_{k+1}$$

In our experiments, the learning rate $\eta_k = 0.005$ and momentum parameter $\beta_k = 0.9$ provided the best results on the given distribution, as in Section 3.3. Momentum helps to mitigate oscillations in directions with high curvature, stabilizing the optimization trajectory and leading to faster convergence compared to gradient descent.

## B.3    MEMFORMERS VS. NAG AND MGD

In our experiments, we observed that Memformers (20) outperform both NAG and MGD on non-isotropic data. Figures 6a and 6b compare the performance of Memformer with NAG and MGD, respectively, on the same non-isotropic data. As shown, the Memformer achieves faster convergence and much better loss performance compared to both algorithms.

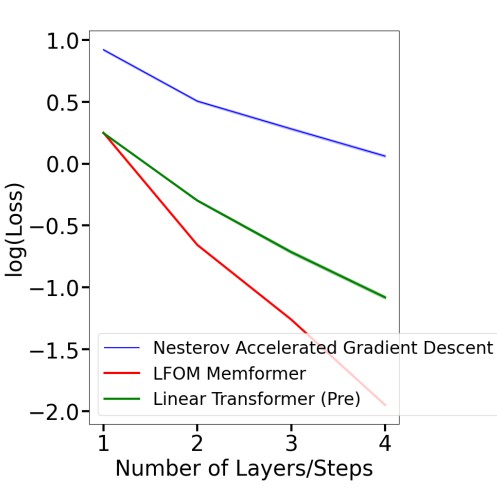 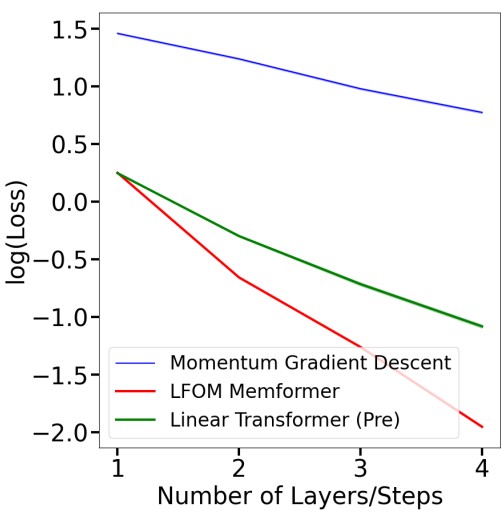

(a) Nesterov AGM vs. LFOM Memformer on non-isotropic data.

(b) Momentum GD vs. LFOM Memformer on non-isotropic data.

Figure 6: Comparison of Nesterov Accelerated Gradient Method (left) and Momentum Gradient Descent (right) vs. LFOM Memformer on non-isotropic data.

## C    MEMFORMER EXPERIMENTS WITH MORE THAN 4 LAYERS

In our experiments, we observed that Memformers with more than 4 layers continue to demonstrate impressive performance in learning optimization strategies. We conducted experiments with Memformers having up to 7 layers and dimension $d = 10$. Training beyond this point becomes impractical due to extensive iteration requirements and significant convergence times, which can span several hours. This limitation is a consequence of computational constraints (e.g., available GPUs) rather than any inherent deficiency of the Memformer architecture itself.

Here, $d$ refers to the rank of the square matrix $\mathbf{X}\mathbf{X}^T$ in the empirical loss quadratic as described in Equation 12.

1. **Experiment 7a** (Dimension $d = 5$, Layers = 5): As expected, Conjugate Gradient Descent (CGD) converges within $d$ steps due to the dimensionality constraint. Remarkably, even though the Memformer only learns general parameters $\mathbf{A}_\ell$ (Equation 9) and $\Gamma_\ell$ (Equation 20), it manages to keep up with CGD for up to 4 steps, showcasing its efficiency.

2. **Experiment 7b** (Dimension $d = 10$, Layers = 7): In this case, CGD does not converge until beyond 7 steps, which aligns with theoretical expectations. Nevertheless, the Memformer remains highly competitive, matching CGD's performance for 6 steps and even performing comparably at 7 steps. This demonstrates the Memformer's robust generalization capabilities, even under more complex conditions.

## D    EXPERIMENT ON CONVERGENCE VERIFICATION FOR MEMFORMER
   ## PARAMETER $\mathbf{A}_\ell$ TO $\Sigma$

Our strategy to train the Memformer (20) was to first train the $A_\ell$'s (9) in each layer $\ell$ on the training batch and then to "fine-tune" the $\Gamma_\ell$'s on the training batch. Therefore, we present here an empirical verification of our results per **Theorem 3** in Ahn et al. (2024).

**Theorem 3.** (**Ahn et al. (2024)**) *Assume that $x^{(i)} \overset{iid}{\sim} \mathcal{N}(0, \Sigma)$ and $w_x \sim \mathcal{N}(0, \Sigma^{-1})$, for $i = 1, \ldots, n$, and for some $\Sigma \succ 0$. Consider the optimization of in-context loss (8) for a $k$-layer transformer with the parameter configuration in Eq. (9) given by:*

$$\min_{\{A_\ell\}_{\ell=0}^{L-1}} f(A).$$

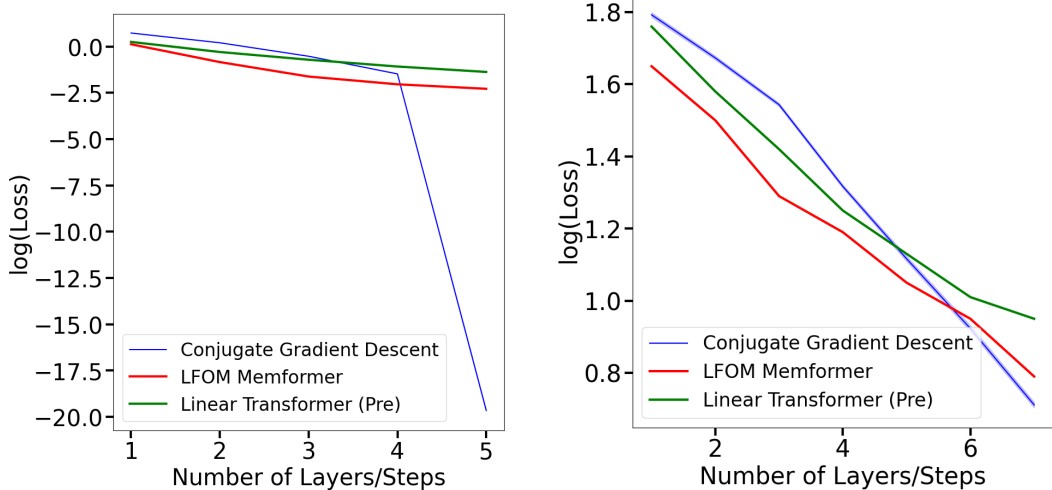

(a) Memformer performance for $d = 5$ with 5 layers.    (b) Memformer performance for $d = 10$ with 7 layers.

Figure 7: Performance comparison of Memformers with CGD for various dimensions and layer configurations.

*Let $S \subset \mathbb{R}^{L \times d \times d}$ be defined as follows: $A \in S$ if and only if for all $i = 0, \ldots, L - 1$, there exist scalars $a_i \in \mathbb{R}$ such that $A_i = a_i \Sigma^{-1}$. Then*

$$\inf_{(A,B) \in S} \sum_{i=0}^{L-1} \|\nabla_{A_i} f(A, B)\|_F^2 = 0,$$

*where $\nabla_{A_i} f$ denotes the derivative with respect to the Frobenius norm $\|A_i\|_F$.*

We evaluated the in-context learning (ICL) loss for linear regression with $d = 5$ and $n = 20$, where $x^{(i)} \sim \mathcal{N}(0, \Sigma)$ and $w_x \sim \mathcal{N}(0, \Sigma^{-1})$. The covariance $\Sigma$ was generated as $\Sigma = U^T D U$, with $U$ being a random orthogonal matrix and $D = \text{diag}(1, 1, 1/4, 1/16, 1)$. A three-layer linear transformer was trained using ADAM, with $A_0, A_1, A_2$ initialized as i.i.d. Gaussian matrices. Each gradient step used minibatches of size 20,000, resampled every 100 steps, and gradients were clipped to 0.01. Results were averaged over 5 runs with independent $U$ and $\Sigma$ samples.

To measure convergence, we computed the normalized Frobenius norm distance:

$$\text{Dist}(M, I) := \min_{\alpha} \frac{\|M - \alpha I\|_F}{\|M\|_F}, \quad \alpha := \frac{1}{d} \sum_{i=1}^{d} M[i, i],$$

which quantifies the deviation of $M/\|M\|_F$ from a scaled identity. The distance $\text{Dist}(\Sigma^{1/2} A_i \Sigma^{1/2}, I)$, averaged over 5 runs, is shown in Figures 8a, 8b, and 8c as a function of training iterations.

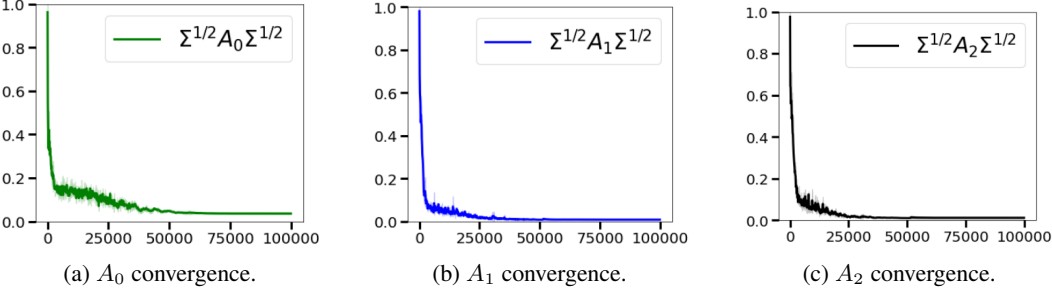

(a) $A_0$ convergence.      (b) $A_1$ convergence.      (c) $A_2$ convergence.

Figure 8: Convergence of $\Sigma^{1/2} A_i \Sigma^{1/2}$ to the scaled identity matrix for each $i$, as predicted by Theorem 3 of Ahn et al. (2024).

