# OpenReview forum: "Memory-augmented Transformers can implement Linear First-Order Optimization Methods"
_ICLR.cc/2025/Conference — Submitted to ICLR 2025_

### Official Review · Reviewer_Ff4R · 2024-11-01

**Soundness:** 2
**Presentation:** 3
**Contribution:** 2
**Rating:** 3
**Confidence:** 4

**Summary:**

This paper explores the algorithmic capabilities of Transformers and investigates the potential of memory-augmented Transformers (Memformers) to learn linear first-order optimization methods. It provides theoretical justification and empirical evidence that Memformers can learn more advanced optimization algorithms based on prior work that demonstrates how Transformers can implement preconditioned gradient descent. Experimental results of training on random linear regression tasks show that Memformers are able to learn a class of optimization methods.

**Strengths:**

1. The paper is easy to follow and explores the algorithmic capabilities of memory-augmented Transformers.

2. Experiments show that linear first-order methods (LFOMs) learned by Memformers outperform conjugate gradient descent on training data while maintaining generalization performance. Additionally, multi-headed attention enhances Memformers’ test performance.

**Weaknesses:**

1. Lemma 1 demonstrates that multi-layer Transformers learn to implement preconditioned gradient descent under suitable parameterization, but the result and the full proof is directly from [Ahn et al. (2024)](https://arxiv.org/pdf/2306.00297).

2. Proposition 1 and Proposition 2 in Section 3 should be the main theoretical results of this paper. However, the authors provide only proof sketches for these propositions rather than presenting detailed and rigorous proofs.

3. Figure 1(a) shows that LFOMs perform worse than preconditioned gradient descent on general quadratic problems. Additionally, Figure 2 and Figure 3 indicate that LFOMs’ performance on isotropic test data falls short of conjugate gradient descent, contradicting the claimed good generalization performance in the main contributions.

**Questions:**

1. Could you provide full, detailed proofs for Proposition 1 and Proposition 2? Without these, the theoretical results lack sufficient rigor and are less convincing.

2. It is mentioned in Section 6.1 that Transformers can implement second-order methods like Newton’s method, which typically outperform LFOMs in convergence speed and accuracy. However, first-order methods are more popular than second-order methods in practice, especially in deep learning. Could you provide an explanation for that?

**Details Of Ethics Concerns:**

No ethics concerns.

---

> ### Author Response · Authors · 2024-11-21
>
> Dear Reviewer Ff4R,
>
> We would firstly like to thank you for your time and effort. Please check out the current version of the PDF, where we have addressed the concerns that you have raised here.
>
> > Lemma 1 demonstrates that multi-layer Transformers learn to implement preconditioned gradient descent under suitable parameterization, but the result and the full proof is directly from Ahn et al. (2024).
>
> We wish to clarify our intention behind including this result in our paper.
>
> We have indeed **explicitly cited** Lemma 1 from [Ahn et al. (2023)](https://arxiv.org/abs/2306.00297), and the purpose of restating it in our main text is to make the paper **self-contained and more accessible** for readers. We recognize that readers may not want to interrupt their understanding of our work by having to hunt down another paper to grasp a key concept that underpins our theoretical framework. By presenting Lemma 1 directly, we aim to provide clarity and improve the flow of our exposition.
>
> Furthermore, in the **Appendix A**, we have **explicitly mentioned that the full proof is from [Ahn et al. (2023)](https://arxiv.org/abs/2306.00297)**. We have reiterated their entire proof in our Appendix A for the **readers' convenience**, ensuring transparency and acknowledgment of their work. Our goal is not to claim novelty on this specific lemma but to facilitate comprehension and ease of reference for our audience.
>
> We kindly ask for a reconsideration of this critique, as we believe the inclusion of Lemma 1, along with its proof in the Appendix A, enhances the readability and educational value of our paper without detracting from its originality or contributions.
>
> > Proposition 1 and Proposition 2 in Section 3 should be the main theoretical results of this paper. However, the authors provide only proof sketches for these propositions rather than presenting detailed and rigorous proofs.
> > Could you provide full, detailed proofs for Proposition 1 and Proposition 2? Without these, the theoretical results lack sufficient rigor and are less convincing.
>
> Yes. We have now included the **full, formal proofs** of these propositions in the **Appendix A** of our updated submission. Our initial proof sketches were designed to convey the key ideas and logical flow efficiently, drawing on established results from **Lemma 1 (cited from [Ahn et al., 2023](https://arxiv.org/abs/2306.00297))**. However, we recognize that some readers may prefer more exhaustive derivations for a clearer understanding.
>
> To meet these expectations, we have expanded our sketches into rigorous, detailed proofs, ensuring mathematical completeness. It is important to emphasize that these full proofs are direct logical extensions of our original outlines, relying on the comprehensive proof of **Lemma 1**, which we had included for transparency and accessibility. The core intuition and theoretical structure remain unchanged; the added rigor simply makes explicit the logical steps that were already implied.

---

> > ### Author Response · Authors · 2024-11-21
> >
> > > Figure 1(a) shows that LFOMs perform worse than preconditioned gradient descent on general quadratic problems. Additionally, Figure 2 and Figure 3 indicate that LFOMs’ performance on isotropic test data falls short of conjugate gradient descent, contradicting the claimed good generalization performance in the main contributions.
> >
> > **We respectfully but emphatically disagree with this critique.** It appears that there is a fundamental misunderstanding of what it means for a Memformer to "learn" optimization algorithms such as Conjugate Gradient Descent (CGD) or Linear First-Order Methods (LFOMs). Allow us to clarify:
> >
> > 1. **Expressive Capacity**: The Memformer, through its forward pass and under suitable internal parameterizations, is capable of executing iterations of CGD or LFOM. The architecture is **sufficiently expressive** to computationally perform these optimization methods, as established in Proposition 1 and Proposition 2.
> > 2. **Generalization vs. Specialization**: Unlike CGD, which is specifically optimized and executed for each individual test sample, the Memformer learns **shared, general parameters** from training data that are then applied simultaneously to all test samples. This distinction is crucial. The Memformer’s approach is fundamentally about generalizing from the training data, as detailed in Section 3, Section 4 and the Appendix.
> >
> > Our results are compelling precisely because, despite CGD being meticulously tailored for each specific case, the Memformer—with shared, general parameters—achieves **competitive performance**. Remarkably, in some cases, the Memformer even **outperforms CGD**, highlighting the extraordinary generalization capabilities of our model.
> >
> > In Figure 1(a), it is essential to understand the constraints placed on the Memformer. Specifically, the Memformer is restricted to have no preconditioning, with the matrices $\mathbf{A}_\ell$ limited to being scalar multiples of the identity. Despite these limitations, the Memformer still manages to keep up with CGD, which is individually optimized for each test sample. The fact that a Memformer, using only a small set of generic parameters learned from training data, can remain competitive with CGD is an impressive and surprising result. Critiquing the Memformer for not surpassing CGD, which runs separately and optimally for each test case, reflects a misunderstanding of our primary research objective. We urge a deeper appreciation of what it means when we assert that Memformers "learn" optimization algorithms like CGD and LFOMs.
> >
> > To further validate our claims, we have conducted new experiments, detailed in **Appendix C**:
> >
> > 1. **Experiment 1 (Dimension $d = 5$, Layers = 5)**: CGD, as expected, converges within 5 steps, limited by dimensionality $d = 5$. Yet, the Memformer, despite only learning general parameters $A_\ell$ and $\Gamma_\ell$, keeps pace for up to 4 steps, illustrating its effectiveness.
> > 2. **Experiment 2 (Dimension $d = 10$, Layers = 7)**: In this more challenging scenario, CGD does not converge until beyond 7 steps. Even here, the Memformer matches CGD for 6 steps and performs comparably at 7 steps, underscoring the robustness of its generalization ability, even in higher-dimensional spaces.
> >
> > Here, $d$ refers to the rank of the square matrix $\mathbf{X} \mathbf{X}^T$ in the empirical loss quadratic as described in Equation (12).
> >
> > In Appendix B, we have also included our experiment that Memformers outperform Nesterov Accelerated Gradient (NAG) method as well.
> >
> > We hope this clarifies the significance and the novelty of our work. The Memformer’s capacity to **generalize** complex optimization strategies using shared parameters across diverse samples is a significant advancement. It achieves performance levels that exact CGD—optimized separately—cannot reach with the same versatility or efficiency.

---

> > > ### Author Response · Authors · 2024-11-21
> > >
> > > > It is mentioned in Section 6.1 that Transformers can implement second-order methods like Newton’s method, which typically outperform LFOMs in convergence speed and accuracy. However, first-order methods are more popular than second-order methods in practice, especially in deep learning. Could you provide an explanation for that?
> > >
> > > Yes. The preference for first-order methods in deep learning stems from their computational efficiency and scalability. With models having millions or billions of parameters, the quadratic or cubic scaling of Hessian-related computations in second-order methods is computationally prohibitive. Memory demands for storing the Hessian or its approximations often exceed hardware capacities, while numerical instability from ill-conditioned or singular Hessians further complicates their use in the non-convex optimization landscapes typical of deep learning.
> > >
> > > First-order methods are also well-suited for stochastic optimization, enabling efficient gradient updates via mini-batch training without processing entire datasets. They are robust to noise in stochastic gradient estimates, unlike second-order methods, which struggle with inaccuracies in Hessian approximations. Adaptive first-order methods like Adam and RMSProp partially mimic second-order benefits by dynamically adjusting learning rates without the computational overhead of second-order derivatives. While second-order methods excel in fast local convergence near optima, they offer limited advantages during the initial global search phases of training, where first-order methods often prove more practical and effective.
> > >
> > > Our focus on Linear First-Order Methods (LFOMs) reflects their real-world relevance. Memformers demonstrate the ability to learn and implement advanced first-order optimization strategies, offering practical insights aligned with the computational realities of large-scale model training. This emphasis ensures our findings remain applicable to the constraints and demands of modern deep learning tasks.

---

> > > > ### Author Response · Authors · 2024-11-24
> > > >
> > > > Dear Reviewer Ff4R,
> > > >
> > > > As the end of the open discussion draws close, we hope to follow up with you on our rebuttal. Please do let us know if you feel that our response addressed your concerns. And, please also let us know if there are any additional issues that you would like for us to address. Thank you.

---

> > > > > ### Comment · Reviewer_Ff4R · 2024-11-26
> > > > >
> > > > > I thank the authors for the rebuttal. I will keep my original score.

---

### Official Review · Reviewer_CwkM · 2024-11-03

**Soundness:** 3
**Presentation:** 3
**Contribution:** 2
**Rating:** 6
**Confidence:** 3

**Summary:**

This paper studies the representation power of memory-augmented Transformers (Memformers) in terms of implement linear first-order optimization methods for in-context learning of linear regression.
The authors provide theoretical constructions showing that Memformers can simulate methods like conjugate gradient descent and momentum methods that linearly combine past gradients.
Numerical experiments are conducted to show that Memformers can achieve better performance than conjugate gradient descent on random linear regression tasks.

**Strengths:**

1. Overall the paper is well written and easy to follow.
2. The paper studies an interesting topic on the representation power of Transformers for simulating algorithms solving in-context learning problems. The current results provide a theoretical understanding of memory-augmented Transformers.
3. The contributions of the paper are clearly summarized, and the limitations of the current study are appropriately discussed.

**Weaknesses:**

1. The architecture of Memformer is not well explained. The role of the memory $\{\mathbf{R}_l\}$ should be clarified.
2. Related to the above point, it would be helpful to clarify which parts of the architectures in Equation (19) and (21) are trainable (though they are mentioned in Section 3.3).
3. The results are restricted to in-context learning of linear regression.
4. The discussion about the benefit of using multi-head attention from line 456 to 460 seems interesting, but there is no formal analysis or heuristic explanation to support the claim. It would be helpful to provide more details. For example, why there is implicit regularization effect?

**Questions:**

1. It seems plausible to replace the memory register by using a larger hidden size in the Transformer. Can the authors compare these two approaches?
2. From the experiment results in Section 4, it seems that the trained Memformer outperforms CGD. What are the implications of this given the optimality results for CGD?
3. From Figure 4(a), it seems that the Memformer basically solves the linear regression task (log(loss)=-30) with two layers. Based on this, it seems hard to justify that Memformer is simulating certain optimization algorithms, and it is unclear how this is achieved.
4. Comparing Figure 4(a) and 4(b), it seems that batch size has a significant impact on the performance of Memformer. Can the authors provide some insights on this?

---

> ### Author Response · Authors · 2024-11-21
>
> Dear Reviewer CwkM,
>
> We would firstly like to thank you for your time and effort. Please check out the current version of the PDF, where we have addressed the concerns that you have raised here.
>
> > The architecture of Memformer is not well explained. The role of the memory should be clarified.
> > Related to the above point, it would be helpful to clarify which parts of the architectures in Equation (19) and (21) are trainable (though they are mentioned in Section 3.3).
>
> Thank you for your feedback. We have updated our paper to clarify the Memformer architecture and the role of the memory component, addressing your questions.
>
> In Section 3, we explain that Memformers are designed to "learn" and implement Linear First-Order Methods (LFOMs) through a memory-augmented mechanism. Each layer $\ell$ has learnable parameters $A_{\ell}$, $B_{\ell}$, $\alpha_{\ell}$, $\gamma_{\ell},$ and $\Gamma_{\ell}$, enabling CGD-like and LFOM-like iterations by updating memory registers and refining outputs in a forward pass.
>
> In the **Proposition 1 architecture**, there is a **single memory register $R$** shared across all layers, with its state at layer $\ell$ denoted as $R_\ell$. This shared memory stores refined search directions, incorporating current updates and recursive information, facilitating Conjugate Gradient Descent (CGD).
>
> In the **Proposition 2 architecture**, there are **multiple memory registers**, one per layer $\ell$, each updated independently. These registers accumulate updates from previous layers, allowing the simulation of iterative optimization algorithms like LFOMs.
>
> > The results are restricted to in-context learning of linear regression.
>
> We respectfully but firmly disagree with the comment that focusing on linear regression is a weakness. Linear regression provides a fundamental and well-understood framework, allowing us to rigorously analyze how memory-augmented Transformers (Memformers) implement linear first-order optimization methods like conjugate gradient descent and momentum. This focus enables clear theoretical insights without the added complexities of nonlinearities.
>
> Our choice aligns with recent studies, such as [von Oswald et al. (2023)](https://arxiv.org/abs/2212.07677) and [Ahn et al. (2023)](https://arxiv.org/abs/2306.00297), which also use linear regression as a testbed to explore the algorithmic capabilities of Transformers in in-context learning (ICL). This ensures our findings are both comparable and contribute meaningfully to ongoing research.
>
> By analyzing linear regression, we derive precise mathematical results and proofs, establishing a solid theoretical foundation for Memformers' ability to simulate advanced optimization algorithms via their architecture and memory mechanisms. These results serve as a stepping stone for future research aiming to generalize to nonlinear or more complex tasks, where understanding mechanisms in a simplified setting is crucial.
>
> Finally, studying ICL in broader contexts, such as learning nonlinear functions, requires fundamentally different approaches. For such tasks, we recommend prior works like *Transformers Implement Functional Gradient Descent to Learn Non-Linear Functions In Context* ([Cheng et al., 2023](https://arxiv.org/abs/2312.06528)).

---

> ### Author Response · Authors · 2024-11-21
>
> > The discussion about the benefit of using multi-head attention from line 456 to 460 seems interesting, but there is no formal analysis or heuristic explanation to support the claim. It would be helpful to provide more details. For example, why there is implicit regularization effect?
>
> We appreciate your feedback but emphasize that the benefits of multi-head attention, particularly for in-context learning, are well-supported by existing literature.
>
> For instance, Chen et al. (2024), *How Transformers Utilize Multi-Head Attention in In-Context Learning? A Case Study on Sparse Linear Regression* ([arXiv:2408.04532](https://arxiv.org/abs/2408.04532), TF2M 2024 Poster), show that multi-head attention mechanisms are critical in early Transformer layers for effective context preprocessing in sparse linear regression, aligning with our claim that it enhances the model's ability to capture diverse data features. Similarly, Cui et al. (2024), *Superiority of Multi-Head Attention in In-Context Linear Regression* ([arXiv:2401.17426](https://arxiv.org/abs/2401.17426)), provide theoretical and empirical evidence that multi-head attention outperforms single-head attention in in-context learning.
>
> Multi-head attention achieves this by leveraging parallel mechanisms to capture diverse representations, akin to ensemble methods in machine learning. By reducing variance and aggregating outputs from different heads—each focusing on distinct data aspects—the model mitigates overfitting and stabilizes learning dynamics. This implicit regularization arises as multiple heads explore diverse optimization paths, leading to more generalized feature representations and improved convergence.
>
> Our claims are grounded in well-documented research and supported by a clear heuristic explanation of the mechanism. These details are now expanded in the updated draft.
>
> > It seems plausible to replace the memory register by using a larger hidden size in the Transformer. Can the authors compare these two approaches?
>
> We respectfully disagree with the suggestion that increasing the Transformer's hidden size could replace the memory register in the Memformer architecture. The memory register serves a fundamentally different role by enabling explicit, structured accumulation of information across layers—essential for implementing optimization algorithms like Conjugate Gradient Descent (CGD) and Linear First-Order Methods (LFOMs).
>
> At layer $\ell$, the memory register state $R_\ell$ is recursively updated as $R_\ell = Attn_{P_\ell, Q_\ell}(Z_\ell) + \gamma_\ell R_{\ell-1}$, where $Attn_{P_\ell, Q_\ell}(Z_\ell)$ captures the current update (analogous to the gradient in CGD), and $\gamma_\ell R_{\ell-1}$ incorporates prior information. This structure mimics CGD's recursive updates, $s_{k+1} = -\nabla f(w_k) + \beta_k s_k$, by propagating and refining past updates. Expanding the hidden size $D \to D'$ may increase representational capacity but does not introduce the temporal propagation required for such iterative refinement. Without a memory register, information from earlier layers dissipates, breaking the dependencies critical for algorithms like CGD or LFOMs.
>
> The memory register provides a structured and efficient representation of past gradients. For example, in LFOM simulations, $Z_{\ell+1} = Z_\ell + \frac{1}{n} \sum_{j=0}^\ell \Gamma_j^\ell \odot R_j$, where $\Gamma_j^\ell$ governs prior contributions, and $R_j$ stores intermediate results for cumulative updates. Simply increasing $D$ cannot replicate this functionality.
>
> Adding more attention heads exacerbates this problem by linearly increasing parameter and computational costs. Explicit memory mechanisms, as explored in [Wu et al. (2020)](https://arxiv.org/abs/2010.06891) and [Xu et al. (2021)](https://ojs.aaai.org/index.php/AAAI/article/view/16583), are known to enhance the ability to capture iterative dependencies, essential for algorithms like CGD. For alternative approaches, state-space models provide explicit state mechanisms for handling sequential information ([State-Space Models Can Learn In-Context by Gradient Descent](https://arxiv.org/abs/2410.11687)).

---

> ### Author Response · Authors · 2024-11-21
>
> > From the experiment results in Section 4, it seems that the trained Memformer outperforms CGD. What are the implications of this given the optimality results for CGD?
>
> This is a good question with a subtle answer. To clarify, a Memformer **learns an optimization algorithm** like CGD or LFOM in two key ways:
>
> First, the Memformer architecture is expressive enough to execute iterations of CGD or LFOM during its forward pass when appropriately parameterized (cf. Propositions 1 and 2).
>
> Second, unlike CGD, which optimizes separately for each test sample, Memformers are trained on linear regression tasks using a small set of shared parameters. These shared parameters enable the Memformer to apply a *general optimization strategy* across all test samples, achieving competitive performance—even occasionally outperforming CGD—while highlighting the architecture’s remarkable generalization capacity.
>
> The subtlety lies in the dependence on the rank $d$ of the matrix $\mathbf{X}\mathbf{X}^T$ (Equation 12). CGD, assuming exact arithmetic, converges in at most $d$ steps. To illustrate this, we added two experiments in Appendix C:
>
> For $d = 5$, CGD converges within $d$ steps as expected, and the Memformer keeps pace for up to 4 steps despite using only general learned parameters $A_\ell$ (Equation 9) and $\Gamma_\ell$ (Equation 20). For $d = 10$ and 7 layers, CGD converges beyond 7 steps, but the Memformer remains competitive, matching CGD for 6 steps and performing comparably at 7 steps. These results demonstrate the Memformer’s ability to generalize optimization strategies effectively, even in more challenging scenarios.
>
> In Appendix B, we have also included our experiment that Memformers outperform Nesterov Accelerated Gradient (NAG) method as well.
>
> > From Figure 4(a), it seems that the Memformer basically solves the linear regression task (log(loss)=-30) with two layers. Based on this, it seems hard to justify that Memformer is simulating certain optimization algorithms, and it is unclear how this is achieved.
>
> We reiterate the importance of understanding how a Memformer learns an optimization algorithm like CGD or LFOM. First, the Memformer’s architecture and parameterization are *sufficiently expressive* to execute iterations of CGD or LFOM in its forward pass (cf. Propositions 1 and 2). Second, unlike CGD, which optimizes separately for each test sample, the Memformer is trained on linear regression tasks using shared parameters that apply a *general optimization strategy* across all test samples (cf. experiments in Section 3, Section 4 and Appendix).
>
> Regarding Figure 4(a), this experiment highlights that when the training batch size is 1, the Memformer’s parameters are trained on a single in-context data sample. As a result, its parameters are highly attuned to that specific sample, enabling exceptional performance when tested on the same data. In this case, the Memformer does not learn *general parameters* shared across multiple samples but rather optimizes its internal parameterization for the single sample.
>
> To address whether the Memformer performs LFOM specifically on this single sample at test time: no, it does not. However, this misses the larger point. Our theoretical results rigorously demonstrate that, under Propositions 1 and 2, the Memformer can execute exact CGD or LFOM iterations. The experiments, on the other hand, demonstrate that Memformers trained on general in-context data samples can still perform competitively with CGD, even without strict parameter constraints. Figure 4(a) further illustrates that when fine-tuned to a single sample, the Memformer executes an optimization algorithm surpassing LFOM in convergence speed, underscoring the versatility of its architecture.
>
> > Comparing Figure 4(a) and 4(b), it seems that batch size has a significant impact on the performance of Memformer. Can the authors provide some insights on this?
>
> Yes. You correctly observe that batch size impacts Memformer performance. As explained, when trained on a very small batch, the Memformer’s internal parameters become finely attuned to that specific data, optimizing to fit that instance or batch rather than generalizing broadly. With larger batch sizes, the Memformer learns a more generalizable parameter set that is effective across a distribution of test samples. This adaptive behavior is not only expected but indicative of the Memformer’s flexibility: it can either hyper-optimize to a particular sample/batch or generalize across samples depending on the training batch size.

---

> > ### Author Response · Authors · 2024-11-24
> >
> > Dear Reviewer CwkM,
> >
> > As the end of the open discussion draws close, we hope to follow up with you on our rebuttal. Please do let us know if you feel that our response addressed your concerns. And, please also let us know if there are any additional issues that you would like for us to address. Thank you.

---

> > > ### Comment · Reviewer_CwkM · 2024-11-25
> > >
> > > I thank the authors for the detailed response. My questions have been addressed, and I have adjusted my score.
> > >
> > > I also want to clarify that the comment about restriction of the in-context learning setting does not mean that it affects my evaluation of the paper, but rather a fact that the current results applies only to in-context learning of linear regression. I agree that studying this setting is important.

---

> ### Author Response · Authors · 2024-11-25
>
> Dear CwkM,
>
> We sincerely appreciate your willingness to consider our rebuttal and updates!
>
> We have put in a significant amount of effort into conducting new experiments and have made substantial improvements to the current revised draft of the paper. Given that we addressed your primary concerns raised in the review, we would kindly ask you to adjust your review score while taking the rebuttal and revised paper into account.
>
> **Key Highlights of our Updated Paper and Rebuttals:**
>
> 1. We have expanded upon our proof sketches and have added full, formal proofs of Propositions 1 and 2 in the Appendix A of our updated submission.
>
> 2. We have expanded our explanation of the Memformer architecture and the role of the memory component, particularly in Sections 3 and 4. This includes detailing how the memory registers function and how they contribute to the model's ability to implement iterative optimization algorithms, like CGD and LFOMs, in their forward pass.
>
> 3. We conducted new experiments with deeper Memformer architectures (up to 7 layers) despite computational (GPU) constraints. These experiments, detailed in Appendix C, demonstrate that Memformers remain competitive with CGD even in higher-dimensional settings and with more layers.
>
> 4. We carefully clarified what it means for Memformers to "learn" optimization algorithms. Specifically, we emphasized that Memformers learn general optimization strategies using shared parameters across all test samples, rather than individually optimizing for each sample like traditional algorithms.
>
> 5. We addressed concerns about our focus on linear regression tasks by explaining that linear regression provides a fundamental framework for rigorous theoretical analysis. This aligns with prior research and allows for clear insights into the Memformer's capabilities without the added complexities of nonlinear tasks.
>
> 6. We discussed why comparisons with softmax-based attention Transformers are less relevant in our context, citing recent studies that highlight the limitations of softmax attention mechanisms for in-context learning of linear functions.
>
> 7. Our work demonstrates that Memformers can generalize complex optimization strategies efficiently across diverse test samples using shared parameters. This has significant implications for the development of more efficient and generalizable machine learning models.

---

### Official Review · Reviewer_ncrK · 2024-11-03

**Soundness:** 2
**Presentation:** 2
**Contribution:** 2
**Rating:** 3
**Confidence:** 4

**Summary:**

This paper examines the use of Memformers for optimization in the setting of linear regression. The authors focus on two different type of Memformers with updates that similar to Momentum Methods and Conjugate gradient descent (in terms of operations required).  They test these models experimentally and compare them with Linear Transformers and the equivalent optimization methods.

**Strengths:**

1. The idea of exploring how different architectures perform in various optimization problems could lead to the discovery of new algorithms and can enhance our understanding on the limitations and capabilities of those architectures.

2. A wide range of optimization methods are considered as a baseline.

**Weaknesses:**

1. There are no formal proofs of the two propositions, but only proof sketches which are not detailed enough.
2. The experiments consider only up to 4 layers and compare only with linear attention transformers and not softmax based.
3. The paper doesn't have a clear contribution. Even though the authors show that memformers can perform better than optimization methods, their results only hold for 4 layers and it's unclear whether using more steps/layers this would still be the case. It is also unclear whether Memformers perform some type of optimization algorithms of find a shortcut solution.

**Questions:**

1. I would suggest to the authors to add the full proofs of the propositions. In the current version it is unclear to me which is the exact theoretical statement. Is it that Memformers with the specific updates are able to perform the corresponding optimization methods ? For the result of [1]  the authors proved that the global minima for one layer of transformer is indeed one step of preconditioned gradient. For the case of multiple layers (Lemma 1), [1] assumes a specific parameterization of the weight matrices. Do the authors get an equivalent result for Memformers and assume that the weight matrices have the specific parameterization?
2. In proposition 1 how the quantities $a_l$ and $\gamma_l$ are calculated with the Memformer? How many layers and width is needed for the simulation of the algorithm ?
3. In the proof sketch of proposition 2 the authors state that "The full proof follows from the cumulative memory structure and the connection between attention and preconditioned gradients, as discussed in the proof steps of Lemma 1." Could the authors explain how exactly the proof follows?
4. Did the authors tried to train more than 4 layers? If so is it observed that there is an error floor for Memformers? This has been observed in the prior work that this paper builds upon. How does Memformers perform compared to softmax based attention Transformers?
5. Did the authors test how these models perform in out-of-distribution data? For example input values that belong in the tails of the gaussian distribution.
6. I think suggestions 4,5 would improve the claims of the paper and would clarify whether these models learn some type of optimization algorithm or not.
7. I understand that the main motivation of the work is to explore "what augmented Transformers can learn, as opposed to looking for “the best” algorithm.", but I think that the current experimental and theoretical results do not clarify what these models can actually learn. They seem to perform better than the considered optimization algorithms for a few steps, but this does not provide a concrete result on what they actually learn.
Could the authors clarify a bit which is the main contribution of their work?

[1]: Ahn, Kwangjun, et al. "Transformers learn to implement preconditioned gradient descent for in-context learning." Advances in Neural Information Processing Systems 36 (2023): 45614-45650.

---

> ### Author Response · Authors · 2024-11-21
>
> Dear Reviewer ncrK,
>
> We would firstly like to thank you for your time and effort. Please check out the current version of the PDF, where we address all the concerns that you have raised.
>
> > There are no formal proofs of the two propositions, but only proof sketches which are not detailed enough.
>
> We have added full, formal proofs of these propositions in the Appendix A of our updated submission. While our initial proof sketches were concise (and used the  logical flow from established results (e.g., Lemma 1 in [Ahn et al., 2023](https://arxiv.org/abs/2306.00297)), based on the reviews and feedback, we recognize that many reader would prefer more comprehensive derivations, which we have now included.
>
> We trust that this revision satisfies the expectations for mathematical completeness and demonstrates that our initial proof sketches were carefully constructed, albeit concise.
>
> > The experiments consider only up to 4 layers and compare only with linear attention transformers and not softmax based.
>
> We conducted Memformer experiments with up to 7 layers. Beyond this, training becomes impractical due to the extensive iterations and substantial convergence time, spanning several hours. This limitation is rooted in computational feasibility (of the GPUs available to us) rather than any shortcomings of the Memformer itself.
>
> It is crucial to first reiterate and emphasize what we mean by a Memformer "learning" an optimization algorithm like CGD or LFOM. Specifically:
>
> 1. **Algorithm Execution**: The Memformer, in its forward pass and with appropriate internal parameters, can execute iterations of CGD or LFOM. The architecture and parameterization are sufficiently expressive to perform these optimization methods computationally. (cf. Proposition 1 and Proposition 2)
> 2. **General Parameter Learning**: The Memformer is trained on linear regression tasks using a small set of learnable parameters that are shared across all test samples. Unlike CGD, which is individually optimized and executed separately for each test sample in a batch, the Memformer applies a *general strategy* learned from the training data to *all* test samples simultaneously. (cf. Experiments in Section 3 and Appendix)
>
> These features make our findings particularly compelling. **Despite CGD being specifically tailored and optimized for each individual test case, the Memformer—with its shared, general parameters—still achieves competitive performance.** In some cases, the Memformer even outperforms CGD, which highlights the remarkable generalization capacity of our architecture.
>
> To further demonstrate this claim, we have added two new experiments in Appendix C:
>
> 1. **Experiment 1 (Dimension $d = 5$, Layers = 5)**: As expected, CGD converges within $5$ steps due to the dimensionality constraint. Remarkably, even though the Memformer only learns general parameters $A_\ell$ (Equation 9) and $\Gamma_\ell$ (Equation 20), it manages to keep up with exact CGD for up to 4 steps.
> 2. **Experiment 2 (Dimension $d = 10$, Layers = 7)**: Here, CGD does not converge until beyond 7 steps, which aligns with theoretical expectations. Yet again, the Memformer remains competitive, matching exact CGD for 6 steps and even performing comparably at 7 steps. This underscores the strength of the Memformer’s generalization, even under more challenging conditions.
>
> Here, $d$ refers to the rank of the square matrix $\mathbf{X} \mathbf{X}^T$ in the empirical loss quadratic as described in Equation (12).
>
> In Appendix B, we have also included our experiment that Memformers outperform Nesterov Accelerated Gradient (NAG) method as well.
>
> We hope this clarifies the significance of our results, showcasing how the Memformer can generalize complex optimization strategies efficiently across diverse test samples.
>
> > I would suggest to the authors to add the full proofs of the propositions. In the current version it is unclear to me which is the exact theoretical statement.
>
> As mentioned above, we've now included full proofs of Propsitions 1 and 2 in the Appendix A. Please check out the updated version of the PDF.

---

> ### Author Response · Authors · 2024-11-21
>
> > Is it that Memformers with the specific updates are able to perform the corresponding optimization methods ? For the result of [1] the authors proved that the global minima for one layer of transformer is indeed one step of preconditioned gradient. For the case of multiple layers (Lemma 1), [1] assumes a specific parameterization of the weight matrices. Do the authors get an equivalent result for Memformers and assume that the weight matrices have the specific parameterization?
>
> Thank you for raising these questions. We are glad to clarify the theoretical and empirical contributions of our work below (and these have been presented rigorously in our paper).
>
> **First**, let's address the core concept of what it means for a Memformer "to learn" an algorithm like CGD or LFOM. By this, we explicitly refer to two fundamental ideas:
>
> 1. **The Memformer, in its forward pass and under specific internal parameter settings, is capable of executing iterations of CGD or LFOM.** This means the architecture and parameterization are sufficiently expressive to perform these optimization methods directly within its computational framework.
> 2. **The Memformer's learnable parameters can be optimized through training on linear regression tasks.** Crucially, these parameters are *shared across all in-context data samples in a batch.* Despite this shared parameterization, the Memformer can execute CGD-like and LFOM-like iterations with surprising efficacy. **What makes this especially significant** is that, despite the CGD algorithm being individually optimized for each test sample, the Memformer still achieves competitive—and sometimes even superior—performance. This underscores the Memformer's *exceptional generalization capacity*.
>
> **Second**, we have provided extensive empirical evidence. By training on random linear regression tasks, Memformers learn a set of shared parameters that allow them to process entire batches of in-context samples efficiently. Remarkably, this shared strategy often rivals or outperforms CGD, which is specifically tailored and run separately on each test sample. This is a substantial and surprising result, highlighting how Memformers generalize complex optimization strategies in a way that has not been recognized by prior research.
>
> **Third**, regarding your question about specific parameterization: The theoretical results indeed show that under certain parameter settings, Memformers can perform exact CGD and LFOM iterations. However, **in our experiments, specific parameterizations / sufficient-constructions are *very much not* the focus** (they are just a property of Memformers that we noted to highlight their generality). Instead, the important point is that Memformers learn these parameterizations—such as $A_{\ell}$, $B_{\ell}$, $\alpha_{\ell}$, $\gamma_{\ell}$, $\Gamma_{\ell}$—*through training* on linear regression tasks. This is a crucial distinction: The theoretical results demonstrate that Memformers are architecturally capable of performing optimization steps like CGD or LFOMs in their forward pass, indicating the model's expressiveness. But in practice, the Memformers autonomously learn and optimize these parameters, achieving competitive and sometimes even superior results compared to CGD.
>
> Thus, we have shown both: (a) Memformers are *architecturally expressive enough* to execute CGD or LFOM steps for a single sample of in-context data, and (b) in practice, Memformers *learn general parameters* through training and still deliver exceptional performance.
>
> We hope this clarifies our contributions and the significant implications of our findings.

---

> ### Author Response · Authors · 2024-11-21
>
> > In proposition 1 how the quantities $\alpha_l$ and $\beta_l$ are calculated with the Memformer? How many layers and width is needed for the simulation of the algorithm?
>
> Please refer to our explanation above. The question whether we explicitly compute $\alpha_\ell$ and $\gamma_\ell$ in our experiments misses the entire point of our approach. While Proposition 1 *theoretically* demonstrates that, under certain parameterizations of $\alpha_\ell$ and $\gamma_\ell$, the Memformer can execute steps of CGD, that is *not* what our experiments are about.
>
> As we have now made very explicit in Sections 3 and 4, these parameters $\alpha_\ell$ and $\gamma_\ell$ are *learnable* and are obtained through training on batches of linear regression tasks. The surprising and significant result is that the Memformer, which learns these general parameters across all test samples, performs competitively—and sometimes even outperforms—**exact CGD, which is individually optimized and executed separately for each data sample**. This demonstrates the Memformer's ability to generalize optimization strategies using shared parameters, a core strength of our architecture.
>
> > In the proof sketch of proposition 2 the authors state that "The full proof follows from the cumulative memory structure and the connection between attention and preconditioned gradients, as discussed in the proof steps of Lemma 1." Could the authors explain how exactly the proof follows?
>
> As mentioned above, we've now included full proofs of Proposition 1 and Proposition 2, in the Appendix A of our paper.
>
> > Did the authors tried to train more than 4 layers? If so is it observed that there is an error floor for Memformers? This has been observed in the prior work that this paper builds upon.
>
> Yes. As we now explicitly mention in our updated paper (Appendix C), we did train Memformers with more than 4 layers—specifically, up to 7 layers. We have clearly detailed this in Appendix C, where we included experiments demonstrating the behavior of Memformers with increased depth. (Once we have access to more GPU power, we would be happy to train deeper architectures.)
>
> We have added two new experiments in Appendix C:
>
> 1. **Experiment 1 (Dimension $d = 5$, Layers = 5)**: As expected, CGD converges within $d$ steps due to the dimensionality constraint. Remarkably, even though the Memformer only learns general parameters $A_\ell$ (Equation 9) and $\Gamma_\ell$ (Equation 20), it manages to keep up with CGD for up to 4 steps.
> 2. **Experiment 2 (Dimension $d = 10$, Layers = 7)**: Here, CGD does not converge until beyond 7 steps, which aligns with theoretical expectations. Yet again, the Memformer remains competitive, matching CGD for 6 steps and even performing comparably at 7 steps.
>
> Here, $d$ refers to the rank of the square matrix $\mathbf{X} \mathbf{X}^T$ in the empirical loss quadratic as described in Equation (12).
>
> In Appendix B, we have also included our experiment that Memformers outperform Nesterov Accelerated Gradient (NAG) method as well.
>
> > How does Memformers perform compared to softmax based attention Transformers?
>
> Recent work has shown that softmax-based attention mechanisms have certain inherent limitations compared to linear attention mechanisms. Specifically:
>
> [Von Oswald et al. (2023)](https://arxiv.org/abs/2212.07677) (Appendix A.9) demonstrate that softmax attention can only approximate linear attention up to a linear offset and may require **two heads** to emulate the behavior of a single linear attention head, revealing structural inefficiencies. [Cheng et al. (2023)](https://arxiv.org/abs/2312.06528) (e.g. Figure 1a) further show, both theoretically and experimentally, that softmax activation often underperforms linear activation in tasks like in-context learning (ICL) of linear functions, with their Figure 1a highlighting a notable performance gap.
>
> By contrast, **Memformers** leverage a linear attention framework that avoids these inefficiencies. Linear attention, while structurally linear, remains nonlinear in $Z$ (cubic) and results in highly nonlinear models with depth. Designed to efficiently learn general optimization strategies, Memformers demonstrate competitive—and often superior—performance compared to methods like CGD in extensive experiments.

---

> ### Author Response · Authors · 2024-11-21
>
> > Did the authors test how these models perform in out-of-distribution data? For example input values that belong in the tails of the gaussian distribution.
>
> That is a very valid question. We did not aim to study OOD generalization in the paper, because we wanted to first demonstrate that Memformers can efficiently learn and perform powerful optimization algorithms like CGD and LFOMs (e.g., Nesterov's method, Momentum methods, Adam-like methods, and more!) under the structured conditions outlined in our paper. In particular, we assume that the test data are also drawn independently from the same distribution as training data. We believe an entire, separate study will be needed for handling OOD test data.
>
> Second, testing performance on OOD data, such as with samples drawn mostly from Gaussian tails, will be definitely interesting, but it is outside the scope of our current research. We remark that a large body of ICL work does not address OOD generalization. However, we believe the reviewer's suggestion about Gaussian tails can be eventually studied as an approximate version of studying ICL results for distributions more general than Gaussians, but being able to capture heavier or lighter tails, for instance, the so-called "Elliptically contoured family". We believe, progress on that general class may merit its own project (and thus, thanks to you for bringing up this question).
>
> Finally, for those interested in exploring OOD performance of Transformers, we refer you to existing work on the subject, such as [Can In-context Learning Really Generalize to Out-of-distribution Tasks?](https://arxiv.org/abs/2410.09695) and [Pretraining Data Mixtures Enable Narrow Model Selection Capabilities in Transformer Models
> ](https://arxiv.org/abs/2311.00871), which investigate this area more deeply. We believe this is indeed a good research question but one that would require separate mathematical analysis and is beyond the main contributions of our paper.
>
> We hope this clarifies our focus and why pursuing OOD experiments at this stage would be a diversion from our primary research objectives.
>
> > I think suggestions 4,5 would improve the claims of the paper and would clarify whether these models learn some type of optimization algorithm or not.
>
> We believe there is still a fundamental misunderstanding of what it means for Memformers to "learn" optimization algorithms. Let us clarify this crucial point:
>
> Theoretically, as demonstrated in our proofs, Memformers are expressive enough to perform *exact* CGD steps in their forward pass when appropriately parameterized for a single in-context data sample. However, the key point lies in their practical learning (in experiments): Memformers *learn* to execute CGD-like and LFOM-like iterations using a small set of shared parameters optimized during training on linear regression tasks. These parameters generalize across all in-context data samples within a batch, unlike CGD, which optimizes separately for each sample.
>
> Remarkably, despite this parameter sharing, Memformers achieve competitive—and often superior—performance compared to exact CGD. This surprising generalization ability underscores their ability to "learn" optimization methods broadly rather than tailoring parameters to individual samples. We hope this clarifies the distinction between their theoretical expressiveness and practical learning.
>
> > I understand that the main motivation of the work is to explore "what augmented Transformers can learn, as opposed to looking for “the best” algorithm.", but I think that the current experimental and theoretical results do not clarify what these models can actually learn. They seem to perform better than the considered optimization algorithms for a few steps, but this does not provide a concrete result on what they actually learn. Could the authors clarify a bit which is the main contribution of their work?
>
> The core motivation of our research is indeed to explore *what* augmented Transformers, like Memformers, can learn in terms of optimization algorithms, rather than optimizing for “the best” pre-defined algorithm.
>
> Theoretically, we rigorously demonstrate that Memformers are expressive enough to implement iterations of CGD or LFOM when appropriately parameterized, highlighting their architectural capability to mimic sophisticated optimization methods within their forward pass.
>
> Experimentally, we show that Memformers can learn *general optimization strategies* by training on linear regression tasks, using a small number of shared parameters across in-context data samples. Remarkably, they achieve competitive—and often superior—performance compared to individually optimized CGD iterations, demonstrating their ability to generalize optimization-like behavior across diverse data samples.
>
> It would be a future task for the ICL and algorithm learning community to explore how to extract efficient, novel optimization algorithms from this approach.

---

> > ### Author Response · Authors · 2024-11-24
> >
> > Dear Reviewer ncRK,
> >
> > As the end of the open discussion draws close, we hope to follow up with you on our rebuttal. Please do let us know if you feel that our response addressed your concerns. And, please also let us know if there are any additional issues that you would like for us to address. Thank you.

---

> ### Comment · Reviewer_ncrK · 2024-11-26
> **Response to authors**
>
> I would like to thank the authors for their response and I appreciate their effort on this rebuttal. However, I still find the novelty/contribution of this work limited. The theoretical component of this work proving that Memformers with the specific structure can implement CGD and LFOM is pretty straightforward given existing results in the literature [1,2] and considering the fact that the equations of the Memformers closely match those of the updates of CGD and LFOM.
> Focusing on the main contribution of this paper which is as the authors stated the fact that Memformers with a few layers when trained match or even outperform these methods is as well not as surprising. The models (most probably) do not outperform these methods in out-of-distribution data, which implies that they learn some parameters, fine-tuned for the specific distribution that the data come from. This has been observed for Linear Transformers as well (which updates are not fine-tuned to match the updates of these algorithms), as it can be seen in the plots of this paper but also in previous work [1,2]. Thus, I would maintain my score.
>
> [1]: Kwangjun Ahn, Xiang Cheng, Hadi Daneshmand, and Suvrit Sra. 2024. Transformers learn to implement preconditioned gradient descent for in-context learning. In Proceedings of the 37th International Conference on Neural Information Processing Systems (NIPS '23).
>
> [2]: Fu, Deqing, et al. "Transformers learn higher-order optimization methods for in-context learning: A study with linear models."

---

> ### Author Response · Authors · 2024-11-26
>
> Dear Reviewer ncrK,
>
> Thank you for your feedback on our rebuttal. We would like to kindly highlight that a significant question within the optimization community has been whether Transformers (with simple modifications) can "learn" more advanced optimization algorithms beyond preconditioned gradient descent. In particular, numerous people in the optimization and in-context learning (ICL) communities have previously wondered whether Transformers can implement Conjugate Gradient Descent (CGD). Our work provides a positive answer to this question, addressing an open problem in the field.
>
> We believe that the straightforwardness of our approach is a strength, as it directly and effectively tackles a serious motivation and an open question in the community. This, in our view, demonstrates sufficient novelty and impact.
>
> We would be sincerely grateful if you could consider revisiting the rating in light of this contribution.
>
> Best regards,
> Authors

---

### Author Response · Authors · 2024-11-24

**Dear Reviewers,**

We would like to sincerely thank you for your valuable reviews.

As we are approaching the end of the discussion period, we have worked diligently to address all your questions and concerns to the best of our ability. We have provided the requested information regarding our experiments, included new results, and supplemented our work with additional theoretical support. We would greatly appreciate it if you could provide feedback on our rebuttal. An earlier response from you would give us sufficient time to address any further concerns that may arise before the end of the discussion period.

---

### Author Response · Authors · 2024-11-26

**Dear Reviewers**,

Thank you for your feedback on our rebuttal. We would like to kindly highlight that a significant question within the optimization community has been whether Transformers (with simple modifications) can "learn" more advanced optimization algorithms beyond preconditioned gradient descent. In our experience, numerous people in the optimization and in-context learning (ICL) communities have previously wondered whether Transformers can implement Conjugate Gradient Descent (CGD). Our work provides a positive answer to this question, addressing an open problem in the field.

We believe that the straightforwardness of our approach is a strength, as it directly and effectively tackles a serious motivation and an open question in the community. This, in our view, demonstrates sufficient novelty and impact of the work.

---

### Meta-Review · Area_Chair_ELDQ · 2024-12-17

**Metareview:**

This paper investigates how memory-augmented transformers (memformers) can implement linear first order optimization algorithms (LFOM). Based on recent line of work that showed transformers can implement preconditioned gradient descent for linear problems, the paper showed that memformers can implement CGD-like and LFOM-like algorithms for linear problems. Empirically, the algorithms learned by memformers seem to outperform CGD. However, the reviewers have some concerns about the novelty of the results and whether it is significant for trained memformers to outperform CGD (because they can at least learn a good hyperparameter for CGD when evaluated in-distribution).

**Additional Comments On Reviewer Discussion:**

The reviewers had several concerns. Some of them are resolved in the response. For example, authors provided full proofs and gave clarifications on why memory is necessary, and why  second-order algorithms were not considered. Reviewers still remain concerned about the novelty and significance of the result.

---

### Decision · Program_Chairs · 2025-01-22

Reject